# Observed positive feedback between surface ablation and crevasse formation drives glacier acceleration and potential surge

Ugo Nanni [1] ✉, Coline Bouchayer[1,2], Henning Åkesson [1], Pierre-Marie Lefeuvre[3], Erik S. Mannerfelt[1], Andreas Köhler [4], Oliver Gagliardini [5], Jack Kohler[3], Louise S. Schmidt [1], John Hult[1], François Renard[2,6] & Thomas V. Schuler [1]

Sudden glacier acceleration and instability, e.g. surges, strongly influence glacier ice loss. However, lack of in-situ observations of the involved processes hampers our ability to understand, quantify and model such a role. We present an analysis of the initiation of a surge (Kongsvegen glacier, Svalbard), focusing on the interplay between climatic and glacier-specific drivers. We integrate two decades of in-situ observations (GNSS, borehole and surface seismometers) with runoff simulations, and remotely sensed surface-elevation changes. We show that initial glacier thinning led to localized acceleration and crevassing. Then, we show that stronger surface melt enabled meltwater to reach the glacier bed. This input promotes high basal water pressure and glacier sliding, and in turn further surface crevassing. Our observations suggest that this positive feedback leads to the expansion of the initially localized instability. Our findings highlight mechanisms that could trigger glacier instabilities under a warming atmosphere beyond the High Arctic.

Current projections of mass loss from glacier and ice caps focus mainly on changes in surface mass balance, often underestimating how glacier dynamics and associated ice loss will change in a warmer climate[1–4]. In the Arctic, which has warmed about three times as fast as that of the global average since 1979[5], both glacier surface mass balance and dynamic mass loss (frontal retreat, iceberg calving and submarine melting), contribute comparably to total ice loss[6,7]. Although recent efforts have improved the estimation[8] and parametrization of dynamic mass loss in large-scale models[9,10], large uncertainties remain on the evolution of ice loss rates[6,7]. These uncertainties stem largely from dynamic instabilities in glacier systems[7] and their unpredictable nature. Notably, there remains no well-defined upper bound to the spatial or temporal scales over which glacier instabilities can occur. Phenomena like glacier surges (glacier-wide acceleration over one or

several orders of magnitude), ice-stream activation and shut-down[11] or ice-sheet scale Heinrich-events[12], are now approached as manifestations along a continuum of instability, driven by shared processes. Understanding the conditions that favor the initiation and propagation of such instabilities is therefore key for improving our predictions of ice masses globally and of glacier-related hazards. In this study, we focus on glacier surges, a common form of instability in the archipelago of Svalbard in the high Arctic[13,14]. Surges can significantly increase the volume of ice transferred from the accumulation area to the ablation area, enhancing the volume of ice discharged into the ocean. For instance, a single surge event in 2012–2013 from one basin of the Austfonna ice cap doubled Svalbard's annual mass loss[15].

Surges are triggered by changes in basal shear stress or resistance at the ice-bed interface, creating zones of low basal friction that may

[1]Department of Geosciences, University of Oslo, Oslo, Norway. [2]Njord Centre, Departments of Geosciences and Physics, University of Oslo, Oslo, Norway. [3]Norwegian Polar Institute, Tromsø, Norway. [4]NORSAR, Kjeller, Norway. [5]IGE, Université Grenoble Alpes, Université Savoie Mont Blanc, CNRS, IRD, Université Gustave Eiffel, Grenoble, France. [6]ISTerre, Université Grenoble Alpes, Université Savoie Mont Blanc, CNRS, IRD, Université Gustave Eiffel, Grenoble, France. ✉e-mail: ugo.nanni0158@gmail.com

propagate up- and/or down-glacier[16]. In addition to glacier-specific drivers (mainly glacier geometry and bed substrate;[17]), climatic drivers play a key role in glacier surge initiation (mainly air temperature and precipitation;[13]). However, the influence of ongoing climatic trends on surge frequency and periodicity remains uncertain[18]. Persistent negative surface mass balance has been suggested to reduce surge activity by reducing driving stress through glacier thinning[19,20], and by altering basal thermal conditions[21]. In contrast, increasing surface ablation has been linked to enhanced surge likelihood[22,23], as meltwater can access previously inefficient or isolated parts of the subglacial drainage system, increasing basal water pressure, which in turn reduces the ice-bed mechanical coupling and thereby facilitates sliding[24]. Although the modeling community is working toward a unified theory of glacier surges[25,26], in situ observations of surge initiation remain rare. This gap limits our ability to disentangle glacier-specific and climatic drivers, which requires multi-variable observations across both spatial and temporal scales[14,25].

We address this observational gap by conducting in-situ measurements during the initiation of an acceleration event in the upper part of Kongsvegen glacier (78.8°N, 13.3°E, Fig. 1a), which may develop into a glacier-wide surge[14,27]. Kongsvegen is located in Svalbard (Fig. 1), where the mean annual air temperature has increased seven times faster than the global average since 1991[5]. In consequence, over one-sixth of Svalbard's ice volume has already melted since the early 20th century[28]. In response to such atmospheric warming and like its neighboring glaciers, Kongsvegen has experienced a significant increase in surface melt since the 1990s[6,29], leading to an expansion of the ablation zone and an upglacier migration of the Equilibrium Line Altitude (ELA; Fig. 1a, blue lines). Under continued warming, the region may become ice-free within the next 400 years[5]. These rapid and ongoing changes provide a lens into the potential trajectory of other glaciated regions worldwide under future atmospheric warming[5,30].

In this study, we investigate, through a multi-method and multi-temporal analysis, how climatic and glacier-specific drivers have worked in tandem to initiate Kongsvegen's current instability. First, we diagnose the dynamical consequences of climate-driven geometric changes (Fig. 1d, e) using on-ice GNSS measurements and geodetically derived elevation changes (Fig. 2; Methods). In particular, we evaluate the relative contributions of the ice-dynamics components and of the changes in basal water pressure conditions to the observed acceleration. Second, we identify the surface and subglacial processes responsible for these changes using passive seismic records collected at the glacier surface and near its bed (Figs. 3, 4; Methods). Third, we interpret Kongsvegen's recent acceleration as the result of a climatic-induced, self-amplifying mechanism we refer to as a *hydro-mechanical feedback*. Finally, we discuss how this mechanism may contribute to a broader glacier-wide destabilization and contextualize the implications of our findings within a global perspective.

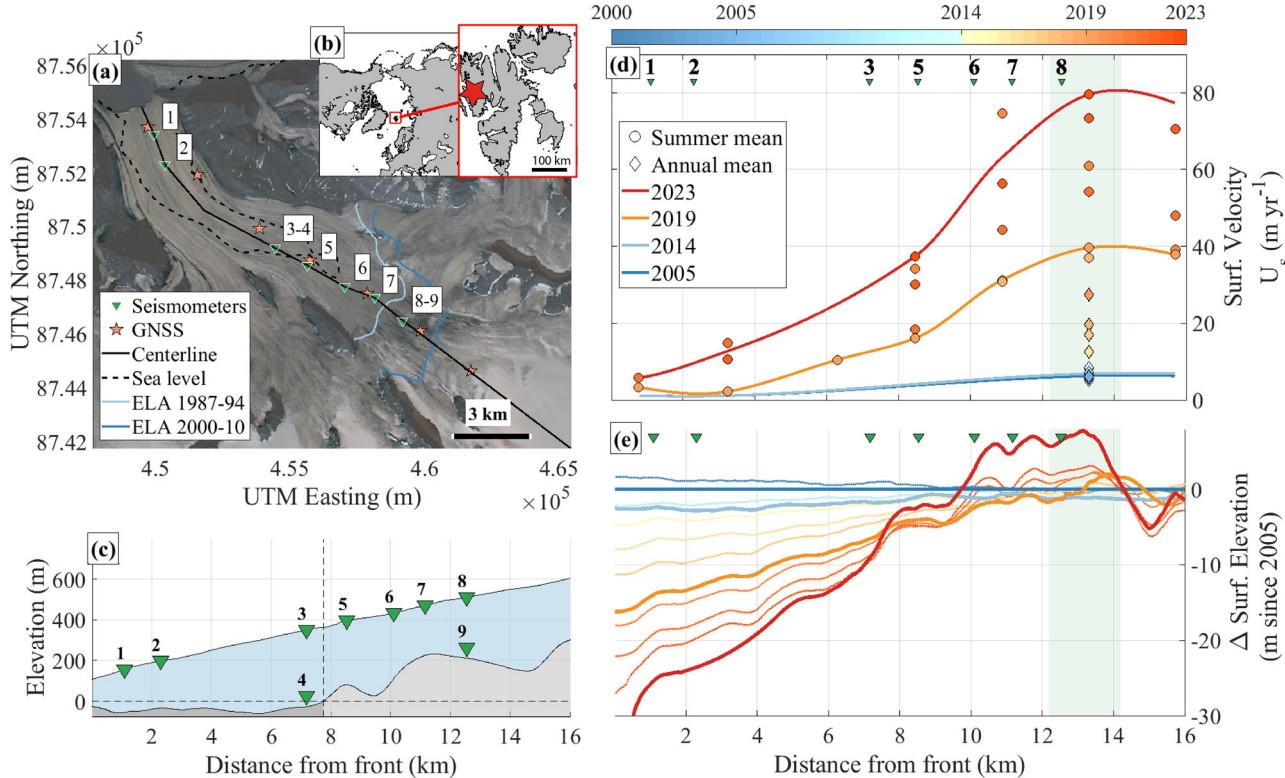

**Fig. 1 | Study area and measurements. a** Kongsvegen glacier (78.8°N, 13.3°E) and the instrumental setup. Green triangles and orange stars show the locations of seismometer groups and GNSS stations, respectively. The black line marks the glacier centerline, the dashed line shows the sea-level elevation at the glacier bed[74], and the blue lines show the long-term equilibrium line altitude (ELA) during the periods 1987–1994 (light blue,[31]) and 2000–2010 (dark blue,[97]). The background image is a Landsat 9 RGB composite taken on August 8th 2023 (https://earthexplorer.usgs.gov/, last access: December 2024. Map services and data available from U.S. Geological Survey, National Geospatial Program.). Inset (**b**) shows the location of the study area (red star) within the Arctic (left) and Svalbard (right). Kongsvegen rests on a 5–60 m thick sediment base[14,98] and has a polythermal regime, with a 50-130 m thick cold layer on top of temperate ice[71]. **c** Surface and bed elevation profiles along the centerline[74] and position of seismometer groups (green triangles). The shaded dark portion of the bed is below sea level. **d** Glacier surface velocity along the centerline with annual velocity (Apr–Apr; diamond markers, 2004 to 2017) and melt-season velocity (Apr–September; circle markers, 2018 to 2023). Solid lines show interpolated velocity profiles for 2005 (dark blue), 2014 (light blue), 2019 (orange), and 2023 (red). **e** Glacier surface elevation changes relative to 2005 based on DEM derived from satellite imagery. **d**, **e** Green shaded area marks the location of the peak of the current acceleration. See Methods for details on the observations. See Supplementary Fig. 1 for the complete timeseries of surface velocity.

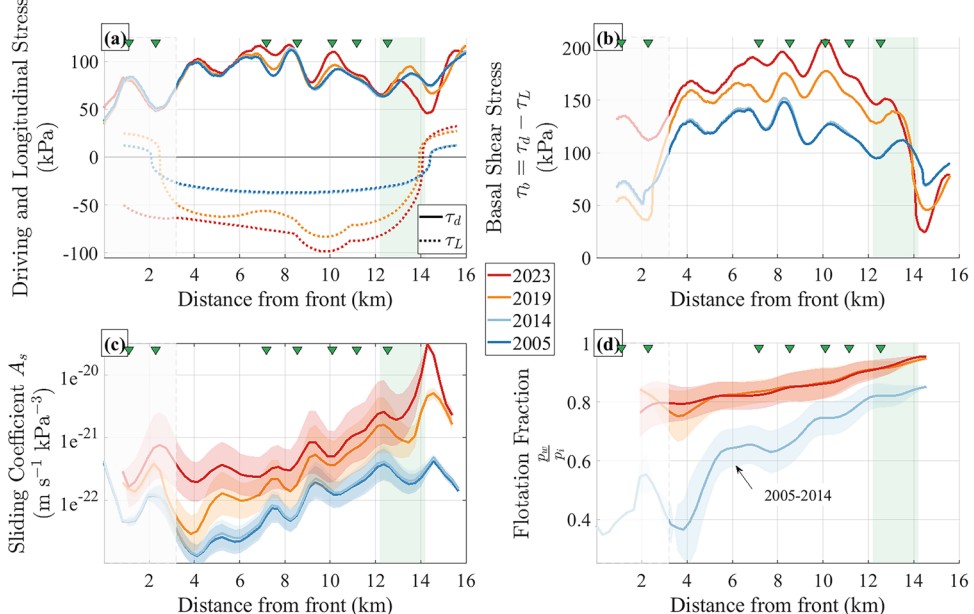

**Fig. 2 | Evolution of glacier dynamics along the glacier centerline.** Years 2005 (dark blue), 2014 (light blue), 2019 (orange) and 2023 (red). (**a**) Driving stress $\tau_d$ (solid) and longitudinal stress $\tau_L$ (dashed). The sign of $\tau_L$ relates to extension (positive) and compression (negative). **b** Basal shear stress $\tau_b = \tau_d - \tau_L$. Note that $y$-axes in (**a**) and (**b**) have common scaling but different offset. **c** Bulk sliding coefficient $A_s$. The blue, orange and red shaded areas represent the envelopes of $A_s$ calculated from $\tau_b = \tau_d - \tau_L$ and $\tau_b = \tau_d$ and the solid line indicates the mean of these bounds. **d** Flotation fractions calculated as the ratio between the basal water pressure $P_w$ and the ice overburden pressure $P_i$. The blue, orange, and red shaded areas represent the envelope of $\frac{P_w}{P_i}$ calculated from $\tau_b = \tau_d - \tau_L$ and $\tau_b = \tau_d$ and the solid line shows the mean of these bounds. Only the averaged conditions are shown for the period 2005-2014. (All panels) Locations of seismometer groups are indicated by green triangles as in Fig. 1. Green shaded area marks the location of the peak of the current acceleration. We do not interpret results within the dimmed area, because $U_s$ changes are too small to allow for a robust stress balance analysis (the limited $U_s$ changes cause unrealistic values of $\tau_L$ and thus $\tau_b$. See Methods).

## Results

### Glacier surface velocity and elevation changes

Kongsvegen glacier experienced a surge in 1948[31] and has since exhibited a slow and stable flow until 2014[14], with mass loss dominated by surface mass balance and little to no influence of calving[32]. From 2005 to 2014 (Fig. 1d; blue lines), Kongsvegen exhibited uniformly low glacier surface velocity ($U_s$) with annual values below 5 m yr$^{-1}$ (Fig. 1d; blue line) alongside noticeable thinning (Fig. 1e; blue line), particularly near the glacier front (up to −5 m from 2005 to 2014). Since 2014 (Fig. 1d; orange and red lines), Kongsvegen exhibits a significant and localized acceleration and continued thinning. While thinning remains most pronounced near the glacier front, the acceleration occurs in the upper part of the ablation zone (locations 7, 8; km 12–14).

In 2023, the lower part of the glacier (locations 1 to 3; km 0–6 Fig. 1d) exhibited low $U_s$, with melt-season (April–September) averages below 20 m yr$^{-1}$. There, thinning reached up to 30 m between 2005 and 2023, mostly related to changes in surface mass balance[32]. Conversely, in the upper part of the glacier (locations 7 to 8; km 12–14) and near the ELA, $U_s$ exhibited a ten-fold increase over the same period, with melt-season averages up to 80 m yr$^{-1}$ in 2023. This area experienced a thickening since 2019, particularly pronounced in 2023 (up to +10 m), while the area immediately up-glacier (km 14–16) displayed minor thinning (>−5 m) and less pronounced acceleration. This contrasting pattern in surface elevation change highlights the formation of a bulge, i.e., a localized thickening, which has expanded and propagated down-glacier between 2019 and 2023[27].

### Stress balance and basal conditions analysis

To investigate the acceleration of Kongsvegen and provide a multi-year perspective on the involved glacier mechanics, we analyze the spatial and temporal changes in its stress balance and basal conditions (Fig. 2a, b). Specifically, we compute the basal shear stress $\tau_b$ and infer key basal friction parameters and relative basal water pressure conditions for four time-slices: 2005 (first $U_s$ measurement), 2014 (onset of the acceleration), 2019, and 2023. These variables are critical for understanding the initiation and propagation of glacier instabilities[25,33].

First, we compute a stress balance diagnosis along a two-dimensional flowline geometry (Eq. (1); Methods), assuming that the driving stress ($\tau_d$) is balanced by basal stress ($\tau_b$), longitudinal stress ($\tau_L$) and lateral stress ($\tau_w$), with

$$\tau_b = \tau_d - \tau_w - \tau_L. \tag{1}$$

We calculate $\tau_d = \rho_i g H \sin(\alpha)$, where $\rho_i$ is the density of ice, $g$ the gravitational acceleration, $H$ the glacier thickness, and $\alpha$ the glacier surface slope. Given the large width-to-thickness ratio of the glacier, we neglect $\tau_w$ (Methods). We estimate an upper bound for $\tau_L$ by considering the surface longitudinal strain rate (Fig. 1d; Methods). In down-glacier direction, $\tau_L < 0$ indicates along-flow compression and $\tau_L > 0$ indicates along-flow extension. Finally, we combine $\tau_d$ and $\tau_L$ to provide a likely upper bound for $\tau_b$.

From 2005 to 2014, neither $\tau_d$ nor $\tau_L$ significantly changed, with average values of $\tau_d$ remaining at c. 75 kPa (solid lines in Fig. 2a) and $\tau_L$ at c. −30 kPa (dashed lines in Fig. 2a). Between 2014 and 2023, $\tau_d$ exhibited a slight increase of up to 10 kPa in the central glacier region (km 6–11), while it decreased by up to 20 kPa in the upper ablation zone (km 13–15). Conversely, along-flow compression ($\tau_L < 0$) intensified in most of the ablation area (km 3–14) over the last decade, with $\tau_L$ decreasing from values near −40 kPa pre-2014 to nearly −100 kPa in 2023. Near the glacier front, the increase in along-flow compression (km 0–3, Fig. 2a) indicates a stagnation of the glacier front, which suggests that there is little to none reduction in frontal stresses due to frontal ablation. In the upper part of the ablation zone (km 14–16), beyond the location of the peak $U_s$ (green shaded area), the glacier

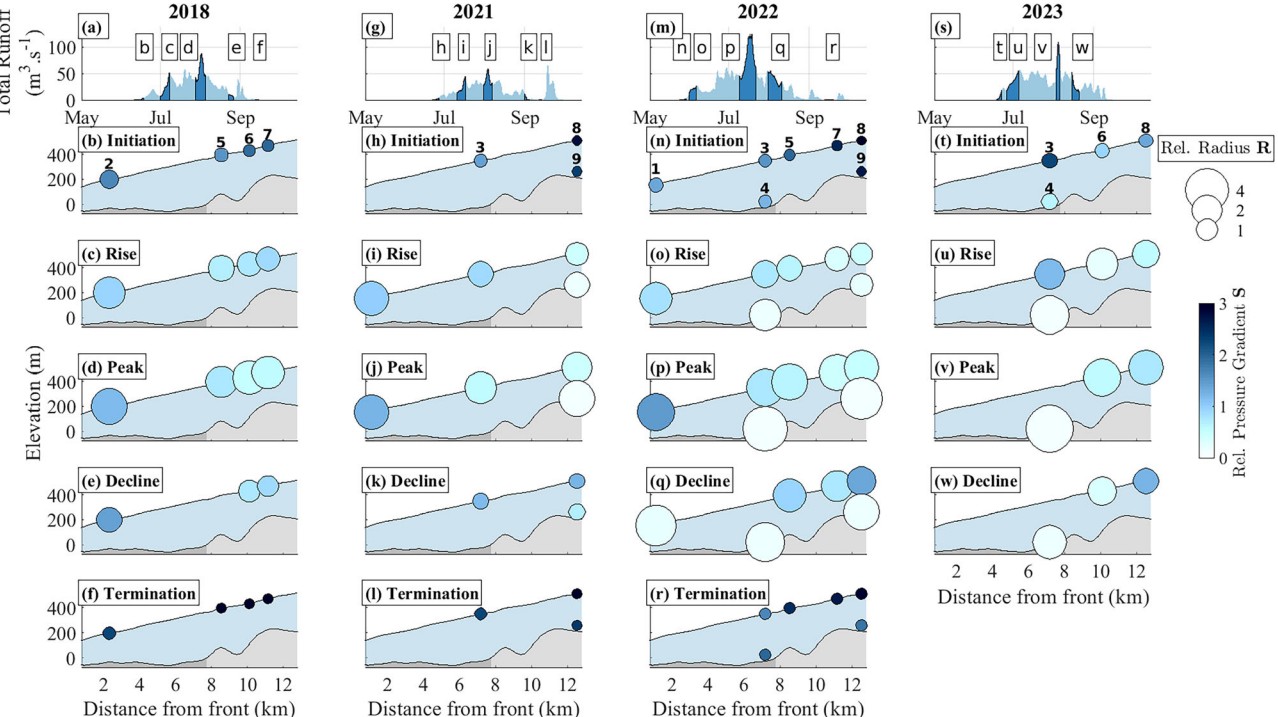

**Fig. 3 | Spatio-temporal evolution of subglacial hydraulic properties.** Each property is shown at the measurement locations (location number shown on top panel as in Fig. 1b). **a, g, m, s** Runoff at location 1 with five highlighted periods (dark blue) detailed below. The periods are chosen to capture the initiation, rise, peak, decline and termination of each of the melt seasons. **b–f, h–l, n–r, t–w** Relative hydraulic radius (size-coded) and relative hydraulic gradient (color-coded) averaged for each period. Shaded areas represent the glacier geometry (blue) over the bedrock (gray). The dark-shaded gray area of the bed shows the glacier part grounded below sea level, and the light-shaded gray area shows the glacier part grounded above sea level. Because of instrument deployment and availability, not all seismometers groups cover the entire period (Table 1). See Supplementary Figs. 2 and 3 for complete time series of runoff and seismic power.

exhibited extensional flow ($\tau_L > 0$), with $\tau_L$ increasing from c. 10 to 25 kPa. Consequently, $\tau_b$ increased from c. 120 kPa in 2014 to c. 180 kPa in 2023 (Fig. 2b; km 3–14) over most of the ablation area and decreased in the upper part of the ablation area from c. 75 kPa in 2014 to c. 25 kPa in 2023 (Fig. 2b; km 14–16).

Second, we retrieve the evolution of Kongsvegen's basal friction conditions from the reconstructed $\tau_b$. We start by reconstructing the evolution of $A_s$, a bulk basal sliding parameter; the higher $A_s$ the more the glacier slips (Fig. 2c). This parameter accounts for various potential basal processes, such as sediment deformation and basal hydrology effects (e.g. basal water pressure). To calculate $A_s$ we use a simple[34]-type sliding law, which can be considered agnostic to bed type:

$$A_s\tau_b^n = U_b. \qquad (2)$$

Here, we use $n = 3$ and derive the basal sliding velocity $U_b$ by calculating the difference between $U_s$ and the glacier's deformation velocity (Methods).

Along the glacier centerline (Fig. 2c), $A_s$ exhibits a persistent spatial pattern through time with a one order of magnitude increase from the lower to the upper parts of Kongsvegen. This heterogeneity likely reflects alternating high and low friction areas at the base of the glacier, potentially due to differences in roughness and/or substrate type. While $A_s$ remained at the same level between 2005 and 2014, it increased significantly between 2014 and 2023. Specifically, $A_s$ increased by one order of magnitude, reaching $2 \times 10^{-20}$ m s⁻¹ kPa⁻³ by 2023 in the area of peak $U_s$ (km 12–16). This increase holds both for the conservative assumption $\tau_b = \tau_d$ (upper limit of the shaded area in Fig. 2c) but also when considering the upper bound of $\tau_b = \tau_d - \tau_L$ (lower limit of the shaded area in Fig. 2c). Although the evolution of $A_s$ highlights a reduction in basal friction since 2014, the formulation of[34]

does not allow distinguishing the influences of substrate and those of hydraulic conditions.

To disentangle these effects, we adopt the approach of ref. 33 and implement a water pressure-dependent sliding law that explicitly accounts for bed friction properties.[35] have proposed a regularized coulomb sliding law appropriate for till bed mechanics that has a similar formulation as the one proposed by refs. 36 and 37 for hard bed mechanics. Considering $C$ to be a constant related to bed roughness and $A$ a bed friction coefficient, we use the following formulation to diagnose the evolution of Kongsvegen's basal water pressure conditions:

$$\tau_b = CN\left(\frac{U_b}{U_b + AC^nN^n}\right)^{1/n}. \qquad (3)$$

Where $N$ the effective pressure difference between the ice overburden pressure $P_i = \rho_igH$ and the basal water pressure $P_w$) and the transition. Here, we assume a uniform distribution of $C = 0.3$, a value likely expected for sediment-type bed[25,37] and we assume a temporally constant $A = \overline{A}_{s(2005, 2014)}$ to reflect the spatial variability of bed properties[33]. Using the inferred $\tau_b$, we derive values of $N$ and subsequently $P_w$ (Methods). Figure 2d illustrates the flotation fraction, $\frac{P_w}{P_i}$, where higher values signify reduced ice-bed coupling.

This procedure translates the variations in $A_s$ shown in Fig. 2c to changes in $\frac{P_w}{P_i}$. For the 2005-2014 period, $\frac{P_w}{P_i}$ shows an increase along the glacier centerline, ascending from the front, with peak values in the upper part reaching 0.85. However, since 2019, $\frac{P_w}{P_i}$ is more spatially uniform, with values exceeding 0.9 across large sections of the centerline. Notably, in 2023, the area experiencing maximum $U_s$ (km 12–16) reached pressure conditions close to flotation ($\frac{P_w}{P_i} = 1$).

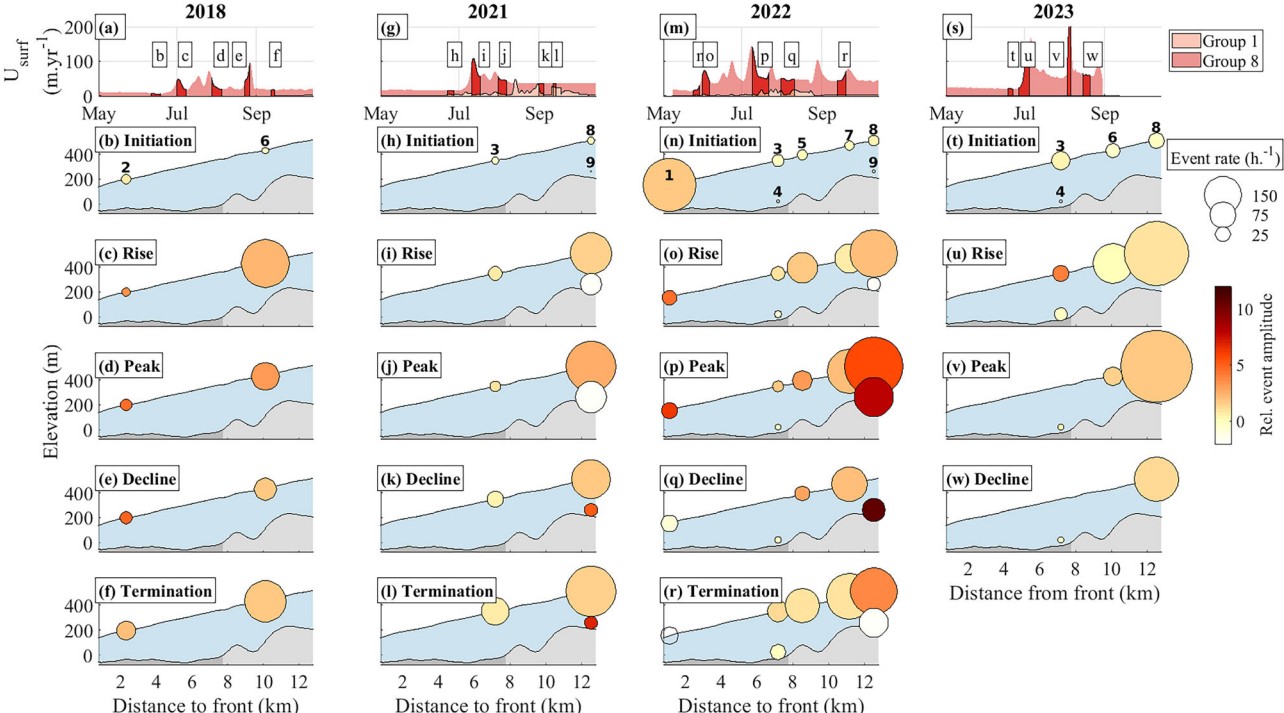

**Fig. 4 | Spatio-temporal evolution of icequake activity.** Each property is shown at the measurement locations (location number shown on top panel as in Fig. 1b). **a**, **g**, **m**, **s** Surface velocity at locations 1 (down-glacier, salmon) and 8 (up-glacier, light red) with the same five highlighted periods (dark red) as in Fig. 3. **b–f**, **h–l**, **n–r**, **t–w** Averaged icequake rate (size-coded) and normalized icequake amplitude (color-coded). Shaded areas represent the glacier geometry (blue) over the bedrock (gray). The dark-shaded gray area of the bed shows the glacier part grounded below sea level and the light-shaded gray area, the glacier part grounded above sea level. See Supplementary Figs. 4 and 5 for complete time series of ice-quake rate and amplitude.

## Seismic investigation

Our diagnosis of mechanical conditions suggests that the recent acceleration of Kongsvegen is primarily due to a decrease in basal friction caused by an increased basal water pressure, particularly pronounced in the upper part of the ablation area (Fig. 2). To unravel the processes responsible for this pressurization of the ice-bed inter-face, we used independent seismic measurements to reconstruct the evolutions of subglacial drainage system properties (Fig. 3) as well as of the occurrence of icequakes to investigate the evolution of crevasses and basal friction (Fig. 4).

Between 2018 and 2023, we operated a network of up to 20 seismometers along the glacier centerline, from the surface to the bed. To ensure data continuity and quality, we grouped the seismometer records into nine distinct groups of closely located instruments (green triangles; Fig. 1a, c; Methods).

We analyzed the subglacial hydraulic properties at each seism-ometer group using the continuous seismic signal. Our focus is on the vertical component of the ground velocity within the frequency band 5–10 Hz (Methods), a range typically dominated by tremor induced by turbulent water flow[38], which can originate, on glaciers, from both an efficient/localized and inefficient/distributed subglacial drainage system[39]. We adopted the framework of[38], based on a forward model relating relative changes in seismic power and subglacial runoff to relative changes in spatially averaged subglacial hydraulic radius and pressure gradient (Methods, Eq. (17)). The hydraulic radius expresses the influence of wall friction for a given cross-section (Methods, Eq. (14)), such that an increase in relative hydraulic radius ($R$; size-coded in Fig. 3) can be interpreted as an increase in subglacial drainage capacity[40]. The hydraulic gradient combines the gradient of subglacial water pressure and the bed slope (Methods, Eq. (15)), such that an increase in the relative hydraulic gradient ($S$; color-coded in Fig. 3) can be interpreted as a pressurization of the subglacial drainage system[38].

Here, we retrieved changes in subglacial hydraulic properties relative to a reference state, defined as June 17th 2018 between 00:00 and 18:00 UTC at location 2.

We analyzed the icequake activity at each seismometer by detecting impulsive seismic events from the vertical component of the ground velocity within the frequency band 25–100 Hz. We continuously compared the short-time average (STA; 0.25 s) of the absolute amplitude of the ground velocity with a long-time average (LTA; 5 s) and identified an icequake when the STA/LTA ratio exceeded predefined threshold values (Methods;[41]). We opted for a constant threshold since we did not observe a significant influence of background seismic noise on the icequake detection (Methods, Section "Effect of noise level on icequake detection"). The STA/LTA analysis yielded a catalog of icequakes (rate size-coded in Fig. 4), each characterized by an amplitude (color-coded in Fig. 4). We did not localize these icequakes; instead, we differentiated between near-surface icequake, typically attributed to surface crevassing[42] and near-bed icequakes, typically attributed to basal crevassing and stick-slip events[43] by comparing records from co-located surface and borehole seismometers.

## Differences between surface and borehole seismometers

In our analysis of hydraulic properties, we find that the hydraulic radius is generally higher when inferred from the basal seismic records than from surface records, whereas the inferred hydraulic gradients are generally lower (Fig. 3). This difference indicates that surface measurements slightly underestimate the efficiency of the subglacial drainage system (see the section "Subglacial hydraulic properties and seismic power" for a quantitative analysis).

For icequake activity, we observe that the icequake rates are generally lower when inferred from the basal seismic records than from the surface records, whereas the amplitudes recorded by the

basal seismometers tend to be higher, particularly during and after peak melt-season (Fig. 4k, l, p, q). Specifically, at locations 8 and 9 and during the 2022 and 2023 seasons, the icequakes detected by the surface seismometers are about ten times more frequent (3,647,186) than those detected by the borehole seismometers (346,408). Only 20% of the borehole-detected icequakes are also detected by the surface seismometers. These differences indicate the dominant sensitivity of surface seismometers to near-surface icequakes and of borehole seismometers to near-bed icequakes, similar to observations at other glaciers[42,43].

### Long-term evolution of subglacial hydraulics and icequakes
For the period 2018 to 2023, the subglacial hydraulic properties do not show a distinct multi-year trend (Fig. 3). Conversely, icequake rates gradually increased in the upper part of the glacier (locations 5 to 9), with annual maxima rising from c. 75 events $h^{-1}$ in 2018 and up to 500 events $h^{-1}$ in 2023 (Fig. 4d, j, p, v). This increase coincides with a gradual increase in $U_s$ in this area, from a melt-season average of c. 28 m $yr^{-1}$ in 2018 to more than 70 m $yr^{-1}$ in 2023 (Figs. 1e, 4a, g, m, s).

### Seasonal evolution of subglacial hydraulics and icequakes
Throughout all melt seasons, surface and borehole measurements indicate a similar seasonal evolution of subglacial hydraulic properties (Fig. 3) and icequake activities (Fig. 4). At the beginning of each melt season (Figs. 4, 3; panels b, h, n, t), runoff and $U_s$ values are low (<10 m³ $s^{-1}$, up to 20 m $yr^{-1}$ in 2023 at location 8, respectively). During this period, the subglacial drainage system operates at high pressure (large $S$) with low capacity (small $R$). Concomitantly, both icequake amplitude and rates (up to 25 events $h^{-1}$ in 2023) are low in all locations, except near the glacier front at the beginning of the 2022 melt season (location 1; Fig. 4n).

As the melt seasons progress (Figs. 4, 3; panels c, i, o, u and d, j, p, v), runoff increases (up to 120 m³ $s^{-1}$ in 2022), initially causing a sharp rise in $U_s$ (up to 100 m $yr^{-1}$ in 2023 at location 8) followed by a decrease at peak runoff. This period corresponds to an increase in subglacial drainage capacity ($R$ up to five times larger than at the onset of the melt season), and a reduction in subglacial water pressure (small $S$ across most of the glacier (locations 3 to 9). However, near the glacier front (locations 1, 2), the subglacial drainage remains pressurized (large $S$). Icequake activity patterns are contrasted along the glacier centerline: regions on a flat bed, grounded below sea level (locations 1 to 4) exhibit low icequake rates (~25 events $h^{-1}$), while regions grounded on a steeper bed above sea level with higher $U_s$ (locations 5 to 9) show significantly increased icequake rates (up to 500 events $h^{-1}$ in 2023) and amplitudes.

Towards the end of the melt season (Figs. 4, 3; panels e, k, q, w and f, l, r), runoff and $U_s$ decrease, although $U_s$ remains at a higher level than its pre-melt-season value. During this period, subglacial pressure increases (larger $S$) and the drainage capacity decreases (smaller $R$), both returning to early season values. In regions grounded on a flat bed below sea level (locations 1 to 4), icequake rate remains low. In regions grounded on a steeper bed above sea level (locations 5 to 9), both icequake rate and amplitude decrease. But by the end of the melt season, icequake rates in these steeper regions are up to four times higher than at its beginning, and icequake amplitudes have increased by one to two orders of magnitude.

## Discussion
Our multi-method observations across spatial and temporal scales provides insights into the processes driving Kongsvegen's recent acceleration. We find that this acceleration results from the interplay of glacier-specific and climatic conditions, ultimately leading to an evolution of the subglacial hydro-mechanical conditions (Fig. 5).

First, we discuss our results to explain the observed acceleration as the outcome of a self-amplifying *hydro-mechanical feedback*. Second, we outline potential near-future pathways for Kongsvegen and

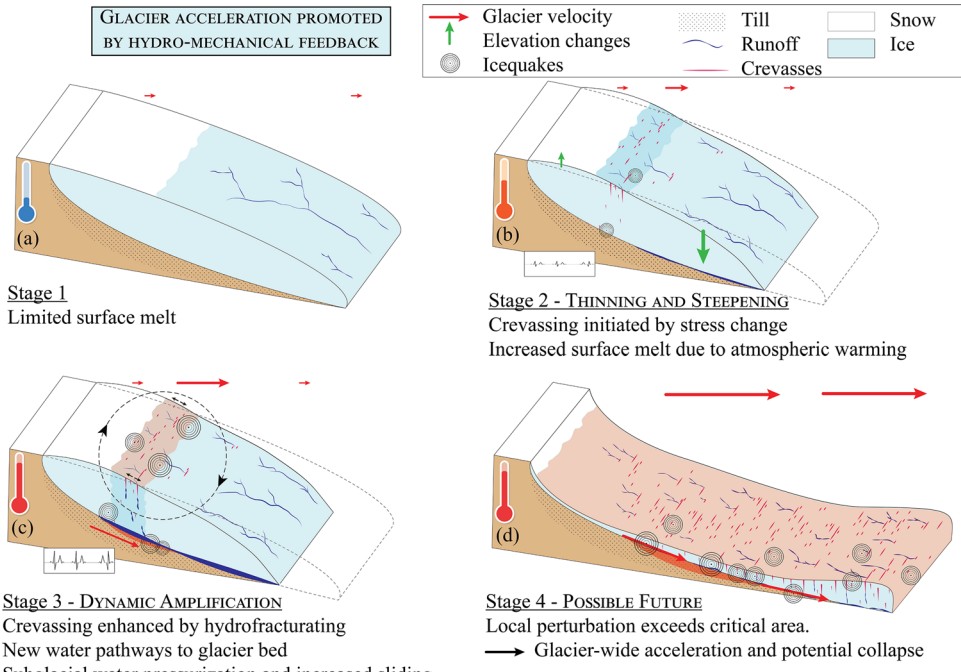

**Fig. 5 | Schematic view of the *hydro-mechanical feedback*. a** Initial state. Limited runoff in the upper part of the glacier and limited meltwater input to the subglacial environment. **b** Stronger surface melt causes the glacier surface to steepen, leading to an initial glacier acceleration and associated extensional flow, which in turn opens crevasses. **c** Crevasses enable distributed routing of water from the surface to the bed. Water reaches previously hydraulically isolated areas, which reduces basal friction. This effect induces further glacier acceleration, even stronger extensional flow and more crevassing, which facilitates a more widespread meltwater supply to the bed, thus perpetuating the positive *hydro-mechanical feedback* loop. **d** Possible future. Glacier wide propagation of the acceleration when the instability reaches a critical length scale, leading to a glacier-wide acceleration and a surge.

identify the triggering mechanisms of its acceleration. Third, we examine the broader implications of this feedback and propose directions for future modeling efforts.

Since 2005, and particularly since 2014, Kongsvegen's lower ablation area steepened (km 0–12; Fig. 1e, f), leading to an increase in driving stress (km 3–12; Fig. 2a). In contrast in the upper ablation zone (km 12–14), since 2014, the glacier slightly thickened and its surface flattened, resulting in a local decrease in driving stress. Concomitantly, surface velocities increased by more than one order of magnitude (Fig. 1d), while the lower area showed only limited acceleration. This dynamic contrast created a zone of enhanced along-flow tensile stress

in the upper part of the glacier (km 14–16; Fig. 1a), favoring crevasse opening[44]. The steady rise in near-surface icequakes activity (Fig. 4) supports this interpretation, pointing to progressive opening of fractures and near-surface crevasses[42,45]. Therefore we suggest that, since 2005 and especially after 2014, surface steepening induced local glacier acceleration, which in turn has promoted the formation of crevasses in the upper part of the glacier (Fig. 5b).

Some newly opened crevasses intersected major supraglacial streams and capture their water (Fig. 6), facilitating water to penetrate deeper into the ice via hydrofracturing, potentially reaching depths exceeding 100 m (≥1/3 of ice thickness;[44]). This mechanism enabled the

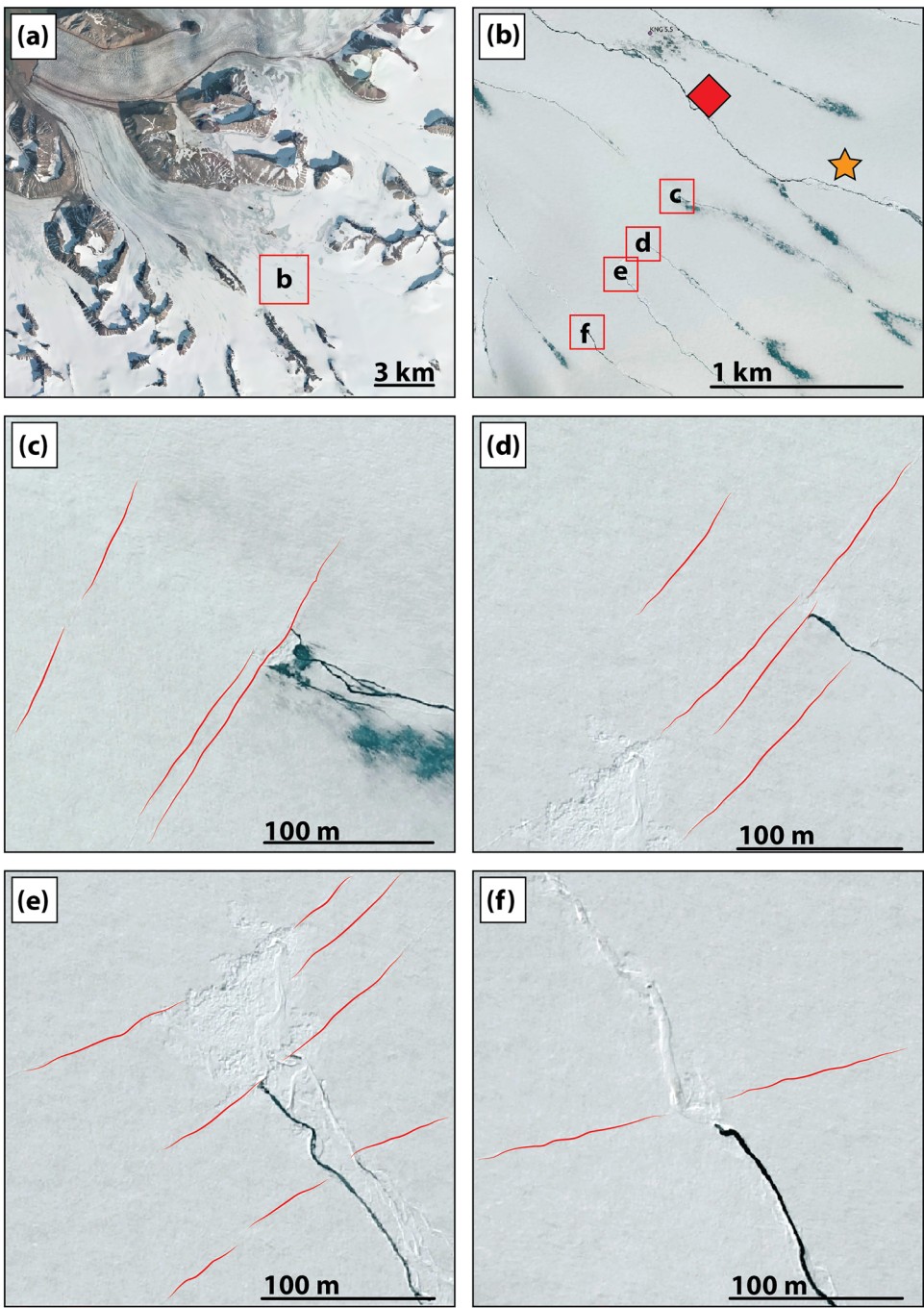

**Fig. 6 | Evidence for supra-glacial streams intersected by crevasses. a** Aerial view on the Kongsvegen on August 1, 2020. **b** Zoom into the upper part of the glacier close to locations 8 and 9 (red square) and the GNSS station (orange star). Note the numerous supraglacial streams. **c–f** insets show supraglacial streams being intersected by crevasses. Note how in (**f**) the intersection seems to be recent since the shape of the supraglacial streams is still visible after the crevasses formed. Basemap from ref. 99.

transfer of surface meltwater to the temperate interior of the glacier, a transfer that previously might have been inhibited by the cold surface layer acting as a thermal seal[46]. Our seismic observations, which show a seasonal evolution of the subglacial drainage system (Fig. 3), support an efficient surface-to-bed connectivity. Therefore, we suggest that crevasse formation has enhanced meltwater supply to the glacier bed (Fig. 5c); consistent with previous studies based on observations[47] and modeling[48,49].

Concurrent with the increase in the surface-to-bed connectivity, we observed both a seasonal control of runoff on glacier velocity (Fig. 4) and a gradual increase in glacier velocity from 2014 to 2023 (Fig. 1d). These behaviors suggest a seasonal modulation of basal friction by surface meltwater[50,51] as well as a multi-year evolution of decreasing basal friction.

This decrease is consistent with Kongsvegen approaching near-flotation conditions in 2023 (Section "Stress balance and basal conditions analysis"; Fig. 2d), and with the observed increase in the rate and amplitude of near bed icequakes (Section "Seismic investigation"). Such icequakes are typically associated with stick-slip and basal crevassing[42,43] and result from variations in basal effective normal stresses in response to changing subglacial water pressure conditions[43,52]. In the case of stick-slip behavior, the increase in icequake rate and amplitude suggests an increase in sliding velocity and/or a decrease in sediment stiffness, indicating a transition to a rate-weakening frictional regime[53]. However, our seismic investigation of subglacial hydraulic properties (Fig. 3) did not capture such a multi-year pressurization. This is likely because the method primarily senses the active, efficient subglacial drainage system[39]. The persistent pressure rise must therefore occur within the inefficient drainage system, as previously observed at this site by ref. 14. Therefore, we propose that the opening of crevasses favors water to reach previously hydraulically isolated parts of the glacier bed, increasing basal water pressure, reducing sediment stiffness, and ultimately enhancing basal sliding (Fig. 5b).

In summary, we interpret the sequence as follows: increasing surface melt leads to glacier surface steepening, initiating glacier acceleration, which induces an increase in along-flow tensile stress, and crevasse opening. These crevasses provide pathways for surface water to the glacier bed. The increased basal water supply reaches previously hydraulically isolated areas, and the associated pressure increase reduces basal friction. This in turn leads to further ice flow acceleration, thereby closing a positive feedback loop (Fig. 5c): the *hydro-mechanical feedback*. This mechanism we describe (Fig. 5) aligns with previous hypotheses for surge initiation in Svalbard[15,22,54]. In the study by ref. 15, surge initiation was facilitated through the transition from a cold to a temperate base due to the supply of meltwater to the bed. On the surging tidewater glaciers studied by refs. 22 and 54, crevasses allowed surface meltwater to access the bed in the lower part of the glacier. However, those studies relied solely on surface and remote-sensing observations, while our analysis integrates in-situ measurements at the bed and surface, which allowed us to disentangle the details of this mechanism.

Between 2019 and 2023, a surface bulge (i.e., local ice thickening) formed and propagated down-glacier (Fig. 1e, km 10–14), accompanied by a reduction in basal shear stress behind this bulge (Fig. 2b) and a down-glacier expansion of the region with increased velocities (Fig. 1d). These features match classic signatures of surge propagation[16,25,27,33]. Although these characteristics indicate that Kongsvegen has initiated a surge, it remains uncertain whether it will (i) remain localized or (ii) evolve into a glacier-wide event (Fig. 5d).

A local and subdued surge (scenario (i)) might arise if stabilization mechanisms dominate. These include: the reduction of basal water pressure through increased basal cavities opening in response to basal slip[55] or through the development of hydraulically efficient pathways to accommodate the increased meltwater supply[56]. In areas where the bed is covered by sediments, increased basal slip could induce sediment dilation, which reduces pore water pressure[57] thereby resisting further glacier motion. These mechanisms could prevent large enough areas of the bed from transitioning into an unstable regime, preventing widespread acceleration.

Alternatively, for a glacier-wide surge (scenario (ii)) to arise, the key condition is that a large enough portion of the glacier bed enters an unstable regime. In[25]'s model and within a rate-and-state friction framework, the initial instability is caused by high basal water pressure, facilitating a reduction in frictional resistance. Our data support such a condition, with our mechanical diagnosis indicating an increase in basal water pressure, and our seismic analysis showing a concomitant increase in basal seismicity, which suggests local rate-weakening regime (Section "Seismic investigation"). Considering current Kongsvegen's glacier dynamics and ongoing atmospheric warming in the Arctic[5], surface melt is likely going to intensify together with surface crevassing. Fueled by the *hydromechanical feedback*, these conditions could lead to a stronger and more widespread meltwater supply to the bed, further expanding the area of instability. However, defining the critical area required for glacier-wide propagation remains challenging due to complex multi-parameter dependencies that are difficult to constrain (Eq. 22 in the Supplementary Information of ref. 25). Regardless of the pathway, Kongsvegen has clearly entered a new regime since 2019, with increased mass transfer from the accumulation to the ablation area, and hence enhanced mass loss.

We propose two approaches to further analyses our observations in order to provide constraints for predictive modeling. On the one hand, a detailed characterization of basal icequakes magnitude and focal mechanisms, as done by ref. 43, to determine spatial extent of the instability. On the other hand, improved constraints on basal shear resistance, particularly distinguishing between ice-bed coupling and till deformation[14]. Such approaches will enable better use of the physical framework of ref. 25 models, ultimately improving predictions of future glacier evolution.

In Kongsvegen, historical surge [[31], 1948] and recent thickening in the accumulation zone[6], suggest a predisposition for dynamic instability. Yet, these mechanical changes alone (i.e., changes in driving stress) cannot explain the observed acceleration without accounting for hydraulically driven reduction in basal friction (Fig. 2). The combination of stronger surface ablation, particularly pronounced towards the front, and an expansion of the ablation zone (Fig. 1) are thus the two main climatic factors that favored the triggering of Kongsvegen's acceleration.

These climatic conditions are now observed for most glaciers worldwide[30]. In particular, thinning of glacier frontal regions and thickening in the upper part of glaciers[58] as well as upward migrations of the ablation zone have been documented across the Arctic, together with evolving subglacial drainage systems[59,60]. While the long-term impact of such a climatic evolution on the total ice loss is still debated[61–63], modeling of Greenland-like synthetic outlet glaciers have shown that this evolution will promote up-glacier expansion of crevassed areas in the near future[48], therefore allowing new meltwater to reach the bed farther inland, potentially triggering similar self-amplified accelerations as observed on Kongsvegen (Fig. 5). For instance, recent collapses of (smaller) glaciers have been linked to comparable crevasse-facilitated increase of surface-base connectivity, such as in the Himalayas[64] or in the European Alps[65].

While our observations are specific to Kongsvegen, the *hydro-mechanical feedback* represents a mechanism that relies on physical processes transferable to other sites. Importantly, our study is conducted on a glacier with a temperate bed, which thus extends the relevance of the mechanism we describe beyond cold-based glaciers compared to previous studies[15]. Our in situ observations thus provide rare empirical evidence on how climate-driven surface processes can trigger dynamic glacier responses leading to a significant short-term increase in ice loss.

At present, the processes responsible for crevasse-facilitated meltwater routing and the resulting increase in basal water pressure and potential trigger of widespread acceleration are barely considered in ice loss and subsequent sea-level-rise projections[1,2,8]. To address this gap, we propose two modeling directions. First, links between ongoing atmospheric warming and the extent and effects of more extensive crevassing needs to be further studied using both observational and theoretical approaches[48]. These conditions could be included in models by explicit calculations of tensile stresses[49,66], or by parametrization of their effects[67]. Second, coupled models of ice dynamics and subglacial drainage need to account for the effect of crevasses on the surface-to-bed hydraulic connectivity[68], for the presence of hydraulically unconnected areas of the bed[69] as well as for mutual effects between sliding and hydraulic conductivity[55]. While in situ data scarcity has largely hindered such progress from being made, our in-situ observations represent an important step toward better representation of glacier instability in predictive glacier models.

## Methods

To investigate the dynamic response of Kongsvegen to a changing geometry and surface melt, we combined seismic measurements, glacier surface-velocity measurements, model-based runoff estimates and derived basal shear stress evolution from glacier surface elevation changes.

### Study area

In the Svalbard archipelago (74°–81° North, 10°–35° East, Fig. 1c), glaciers and ice caps cover 57% of its 60,000 km$^2$ land surface area, corresponding to approximately 6200 km$^3$ of ice[70]. Most of Svalbard's glaciers are polythermal glaciers, composed of a basal temperate layer and an upper cold layer[71]. The climatic mass balance of Svalbard between 2000 and 2019 is estimated at −6.50 ± 3.71 Gt a$^{-1}$, and accounting for frontal ablation in addition, the total mass balance amounts to −8.28 ± 6.05 Gt a$^{-16}$.

We show in Fig. 6 a satellite imagery take on August 1, 2020, showing supra-glacial channels recently cross-cut by crevasses (highlighted in red) in the upper part of Kongsvegen. For all pictures, water flow is from the bottom right to the top left corner, and we can observe that the supra-glacial water flow has been recently interrupted.

### Runoff

We used the runoff from simulations by ref. 29 that used the CryoGrid-community model[72]. Simulations of the surface energy balance and associated snow and ice melt were conducted at 3-h time steps on a 2.5 × 2.5 km grid. Melting was calculated using an energy-balance approach and the model incorporated a percolation scheme to account for refreezing and retention within snow and firn. Horizontal flow between grid points was not considered in the model, but it included a horizontal runoff delay scheme based on the surface slope. At the same locations where our seismometers were installed, we integrated the runoff across their respective hydrological catchment[73,74]. To relate changes in surface runoff and variations in subglacial drainage discharge Q, we assumed that the transfer of surface water to the subglacial environment occurred at sub-daily rate. This assumption is supported by in-situ observations carried out in Svalbard on glaciers with a polythermal regime analogous to that of Kongsvegen[46] and the sensitivity analysis of[29]. [29] found overall satisfactory model performance in terms of surface mass balance and daily discharge at the catchment scale using in-situ observations for a neighboring glacier.

### Glacier surface velocity

From 2018 to 2023, we derived glacier surface velocity from records of four Global Navigation Satellite System (GNSS) stations located in close vicinity (<1 km, Fig. 1a) to our seismic network. The stations collected data at five-second intervals between April 1 and September 1 and operated for one hour per day during the rest of the year. We applied a one-hour static post-processing time window to the GNSS data, which accounts for the relatively low speed of the glacier, and then applied a one-day moving median. We used the Norwegian Mapping Authority permanent network base station in Ny-Ålesund for reference (baseline of ~30 km). For the period from 2005 to 2018, annual glacier surface velocity was derived from annual GNSS surveys of mass balance stakes[32]. When averaging over annual period, this corresponds to April of a given year to April of the following year, when over the melt-season this corresponds to April to September.

### Field equations

**Basal shear stress.** In order to evaluate the temporal and spatial changes in basal shear stress $\tau_b(x, t)$ we follow Eq. 8.15 of ref. 75 and consider the main three components, the driving stress $\tau_d$, the longitudinal stress $\tau_L$ and the lateral stress $\tau_w$ as:

$$\tau_b(x,t) = \tau_d(x,t) - \tau_w(x,t) - \tau_L(x,t). \tag{4}$$

In order to derive $\tau_d$, we follow a one dimensional shallow ice approximation formulation as

$$\tau_d(x,t) = \rho g H(x,t) \sin(\alpha(x,t)), \tag{5}$$

where $\rho$ = 910 kg.m$^{-3}$ is the density of ice, $g$ = 9.81 m.s$^{-2}$ is the gravitational acceleration, $H$ is the glacier thickness, and $\alpha$ is the glacier surface slope. We calculate $\alpha$ using a centered, 1 km along-flow window.

In order to derive $\tau_L$, we follow[76]'s formulation as

$$\dot{\epsilon}_L(x,t) = A_g \tau_L(x,t)^n, \tag{6}$$

where $\dot{\epsilon}_L$ corresponds to the longitudinal strain rate, $A_g$ = 4.4 × 10$^{-25}$ Pa$^{-3}$ s$^{-1}$ the creep factor (for an ice temperature of −9°C;[75]) and $n$ = 3 the creep exponent. Here we evaluate the upper bound of the longitudinal stress, which is obtained at the surface of the glacier where $\epsilon_L = \epsilon_{xx}$ in the case of a confined flow in the across-flow direction[75]. The surface longitudinal strain rate $\dot{\epsilon}_{xx}$ along the glacier centerline can be expressed as:

$$\dot{\epsilon}_{xx}(x,t) = \frac{dU_s(x,t)}{dx}, \tag{7}$$

where $U_s$ is the glacier surface velocity. In order to interpolate the point measurements of $U_s$, we use an Akima piecewise cubic Hermite interpolation[77]. Combining Eqs. (6) and (7) we then obtain

$$\tau_L(x,t) = \left(\frac{\dot{\epsilon}_{xx}(x,t)}{A_g}\right)^{1/n}. \tag{8}$$

Finally, we evaluate $\tau_w$ using the shape factor parameter $f$ so that $\tau_w = (1 - f)\tau_d$ (Eq. 8.16;[75]). Given the geometry of Kongsvegen with an averaged width of 2.5 km and an averaged thickness of 250 m, we obtain a half-width/thickness ratio of 5 and thus a value of $f$ close to 1 (Table 8.5;[75]). We can therefore calculate the driving stress as $\tau_b(x,t) = \tau_d(x,t) - \tau_L(x,t)$.

**Basal sliding parameters.** In order to investigate the changes in basal sliding, we first calculated an effective basal sliding coefficient $A_s$ using[34]' formulation of the sliding law before disentangling into different components using Eq. (11) to derive the effective pressure at the base.

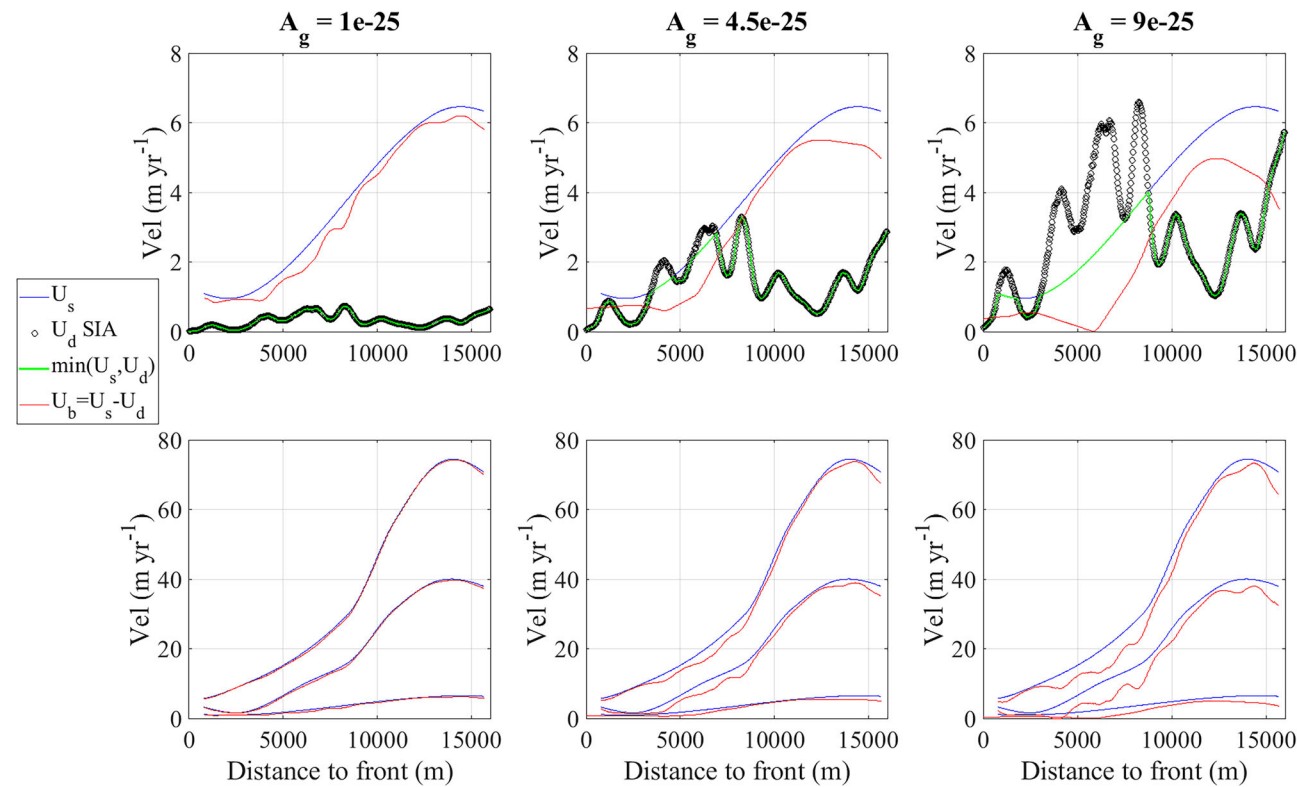

**Fig. 7 | Evaluation of the sensitivity of the deformation velocity to the creep parameter $A_g$ (Pa$^{-3}$ s$^{-1}$).** Each column corresponds to a different value of $A_g$. Upper panels show the measured surface velocity (blue line, $U_s$), the calculated deformation velocity (black dots, $U_d$), the minimum between $U_s$ and $U_d$ (green line) and the calculated basal velocity (red line, $U_b$). Bottom panels show the values of $U_s$ and $U_d$ for the three period of time 2014, 2019 and 2023 as in Fig. 1.

In order to calculate $A_s$ we used[34]' formulation as

$$U_b(x,t) = A_s(x,t)\tau_b(x,t)^n, \tag{9}$$

where $U_b$ is the basal sliding velocity.

Here we calculated $U_b = U_s - U_d$, with

$$U_d(x,t) = \frac{2A_g}{n+1}\tau_b(x)^n H(x,t). \tag{10}$$

We use $A_g = 4.4 \times 10^{-25}$ Pa$^{-3}$ s$^{-1}$ and the sensitivity of $U_d$ to $A_g$ is shown in Fig. 7.

In order to further investigate disentangle $A_s$ in different components, we used[36]' formulation with a post-peak exponent =1[78] as

$$\frac{\tau_b(x,t)}{N(x,t)} = C\left(\frac{U_b(x,t)^{(1-n)}}{C^n N(x,t)^n A(x) + U_b(x,t)}\right)^{1/n} U_b(x,t), \tag{11}$$

where $A$ is the sliding coefficient, $N = p_i - p_w$ is the effective pressure with $p_i = \rho g H$ the ice overburden pressure, and $p_w$ the basal water pressure, and $C$ the maximum value of $\tau_b/N$. In the absence of bedrock roughness data we assumed a uniform and constant roughness with $C = 0.3$[78]. Because of this arbitrary choice of the value of the C parameter, the inferred effective pressures should not be regarded as actual values, and we mainly focus on relative changes. Note that in Eq. (11) $A$ is constant over time. Indeed, as $A$ depends mostly on the substrate roughness, we did not consider such a parameter to significantly change over time and therefore take $A = A_s(x, t = t_0)$ as in[33].

From Eq. (11), we extract $N$ as

$$N(x,t) = \frac{1}{C}\left(\tau_b(x,t)^{-n} - \frac{A(x)}{U_b(x,t)}\right)^{-1/n} \tag{12}$$

by following, with a simplified notation:

$$\frac{\tau}{N} = C\left(\frac{U^{(1-n)}}{C^n N^n A + U}\right)^{1/n} U,$$

$$\left(\frac{\tau}{NCU}\right)^n = \frac{U^{(1-n)}}{C^n N^n A + U},$$

$$\left(\frac{\tau}{NCU}\right)^n C^n N^n A + \left(\frac{\tau}{NCU}\right)^n U = U^{(1-n)},$$

$$\left(\frac{\tau}{U}\right)^n A + \left(\frac{\tau_n}{NC}\right)^n U^{(1-n)} = U^{(1-n)},$$

$$\tau^n\left(U^{-n}A + (NC)^{-n}U^{(1-n)}\right) = U^{(1-n)},$$

$$\tau^n U^{-n}\left(A + (NC)^{-n}U\right) = U^{(1-n)},$$

$$\tau^n\left(A + (NC)^{-n}U\right) = U,$$

$$(NC)^{-n} = \tau^{-n} - \frac{A}{U},$$

$$N = \frac{1}{C}\left(\tau^{-n} - \frac{A}{U}\right)^{-1/n}.$$

Finally, we derived the flotation fraction as the ratio between $p_w$ and $p_i$.

### Seismic data

**Instrumentation.** In 2018, we operated five three-component seismometers positioned along 13 km of the central flowline between the glacier front and the long-term equilibrium line altitude (Fig. 1a). In 2021, this network has been expanded and we operated up to 20 seismometers of which 18 were deployed at the surface (i.e. 1.5 m into the ice to prevent melt-out and loss of coupling during the melt season) and two were installed into boreholes, c. 78 m above the glacier bed (262 m beneath the surface) and c. 86 m above the glacier bed (263

m beneath the surface) (Fig. 1a,b). Surface seismometers were 4.5 Hz sensors and we used HG-6 OB 14 Hz 375 $\Omega$ sensors in the boreholes; data of all sensors were recorded using DiGOS DATA-CUBE³s digitizers sampled at a frequency between 50 and 800 Hz.

To maximize data continuity and quality, we define nine groups of closely located ( < 1 km) seismometers, representing an along-glacier profile (locations 1, 2, 3, 5, 6, 7, 8), complemented with locations at depth (locations 4 and 9; Fig. 1a, b and Table 1). For each of these nine locations, we created a composite record by selecting the best individual record between two field campaigns based on their respective data quality. This approach is similar to the one of[40] for the seismic power and, given the wavelength investigated (150 to 300 m) and the large distance between the different groups ( > 3 km), does not significantly affect our investigation.

**Seismic Power.** We calculated the seismic power $P$ using the vertical component of the ground velocity using Welch's method over a two-second time window with a 50% overlap[79,80] within the frequency band 5 to 10 Hz. Our choice of these frequencies was based on the premise that, within this bandwidth, the signal is dominated by seismic noise induced by turbulent water flow[38,40], whereas higher frequencies are influenced by bedload sediment transport[38]. This has been verified in other glacial settings[81,82]. To maximize sensitivity to continuous background seismic noise while limiting the influence of impulsive short-lived icequakes, we applied a rolling minimum filter to five-minute time windows within the decimal logarithmic space[40].

We show in Fig. 8 the details on how we stacked the records for each group, illustrated with the seismic power. We selected the record with the best quality, and observed that when two records from neighboring stations are available at the same time, they are very similar. This shows that the grouping approach allows us to keep the best data, with a limited influence on the analysis. In Fig. 8, we show the seismic power recorded at the 20 seismic stations, each associated to a group from G1 to G9 as in Table 1. The light colors (salmon and light green) show the available stations/time periods that are not used when grouping the seismometers, while the dark colors show the stations/time periods selected for the grouping. Red colors are for groups with odd numbering, green for groups with even numbering.

**Icequake activity.** At each seismometer, we analyzed the icequake activity by detecting impulsive seismic events. We identified such events by comparing a running window of 0.25 second short-term average (STA) to a five second long-term average (LTA) of the vertical component of the ground velocity between 25 and 50 Hz[41,80]. Event initiation was identified when the STA/LTA ratio exceeds 4.0, while event termination was defined when the ratio equals 2.0. The amplitude of an event was calculated as the 99th percentile of the waveform envelope encompassing the period from 0.5 seconds before event initiation to its termination.

This choice of parameters (i.e., threshold, duration and frequency range) allows detecting short term impulsive events likely originating from brittle or frictional events such as crevasse opening and basal stick-slip related to ice-bed mechanical coupling, while limiting the influence of tremor-like signals produced by calving events and water flow[83]. When such catalogs of icequakes are derived from surface seismometers, they are mostly sensitive to surface crevasses[42], while when derived from borehole seismometers, they are more sensitive to basal icequakes (e.g., basal crevasses and stick-slip events;[43]). This distinction allows us to assess where the majority of seismic signals originates from and to interpret the mechanisms and processes involved.

In order to correct for potential effects specific for individual stations due to installation and location (e.g., local amplification, local seismic activity), we remove the respective winter means from the icequake rate and the seismic power at each station and normalize the icequake amplitude at each station by its respective winter standard deviation.

**Effect of noise level on icequake detection**
We used high-frequency seismic data to derive the characteristics of icequake activity (rate and amplitude). Previous studies[42,84] have shown that icequake detection can be influenced by water-flow induced seismic background noise, such that an increase in noise amplitude may decrease the sensitivity of the STA/LTA trigger algorithm. To investigate the potential effect of background noise level on the icequake detection, we compared the background seismic noise level with the icequake amplitude and detection rate in Fig. 9. We have selected here the seismic power within the bandwidth [5-10] Hz to represent the background seismic noise, because this frequency band dominates the water-flow induced seismicity (Sec. 4). We observe that for seismometers located in the lower part (location 3 - green, Fig. 9a) and upper part of the glacier (location 8 - red, Fig. 9a), the background seismic noise increases during the melt season in response to the increase in runoff. In contrast, the evolution of the icequake rate (Fig. 9b) shows a concomitant stagnation for the lower location (location 3 - green, 9b) and an increase for the upper location (location 8 - red, Fig. 9b). For both locations, these changes in icequake rate are accompanied by increased icequake amplitude (Fig. 9c). Such a contrasting evolution suggests that direct effects (icequake generation rate) and indirect effects (detection rate) of noise levels have little effect on the icequake detection. This observation therefore supports the suitability of our method in such context. While the overall positive correlation or lack of negative correlation between increase in noise and icequake rate proves that changes in the icequake occurrence are not due to changing trigger sensitivity, we do see a few time periods with negative correlation and hence an effect on the sensitivity cannot be excluded at all times.

The adaptive STA/LTA detector by ref. 85 can improve the traditional STA/LTA method by dynamically adjusting the STA/LTA threshold over time based on the varying characteristics of the seismic background noise. This approach allows reducing false detections since it detects only events with STA/LTA values statistically significantly different from the background noise. The noise STA/LTA distribution, for example of a one-hour noise record, is thereby assumed to follow the central F distribution. We tested this method on our data. For time periods with low or moderate icequake activity, the noise STA/LTA distribution can be indeed parametrized by a central F distribution and, thus, we would be able to use adaptive thresholds for producing the event time series. However, during peak activity during melt season, we found that this distribution can no longer be approximated by an F distribution, most likely since the number of events per hour becomes too high, i.e., icequakes are dominating the background wavefield. We therefore decided to keep a consistent constant STA/LTA threshold for all time periods and stations.

**Subglacial hydraulic properties and seismic power**
To investigate the subglacial hydraulic properties, we adopt the framework of[38] based on a forward model of seismicity produced by turbulent flow. Turbulent water flow in a river or a subglacial drainage system generates frictional forces acting on the near boundaries (e.g., river bed or conduit wall), which in turn cause seismic waves with given amplitude and spectral signature. Considering that the main parameters of subglacial water are the subglacial hydraulic radius $R$, the subglacial hydraulic gradient $S$, the number of subglacial channels $N$ and the a parameter $\beta$ accounting for the bed roughness and the channel shape,[38] described the seismic power induced by turbulent water flow $P$ as

$$P \propto N\beta R^{14/3} S^{7/3}. \tag{13}$$

**Table 1 | Latitudes and longitudes of seismic stations deployed on the glacier from 2018 to 2023**

| Stations | Time Period | | | | | | | |
|---|---|---|---|---|---|---|---|---|
| | 2018-05-01 | ... 2020-09-15 | ... 2021-05-03 | ... 2021-08-21 | ... 2022-05-02 | ... 2022-08-29 | ... 2023-04-28 | ... 2023-10-09 |
| KGS00-G1 | | | | (78.844475, 12.66700) | | (78.844475, 12.66700) | | |
| KGS01-G8 | (78.784104, 13.117597) | | (78.784104, 13.117597) | (78.784219, 13.116846) | (78.784219, 13.116846) | (78.784407, 13.115452) | (78.784407, 13.115452) | |
| KGS02-G8 | | (78.782769, 13.120329) | (78.782769, 13.120329) | (78.782884, 13.119577) | (78.782884, 13.119577) | (78.783072, 13.118183) | (78.783072, 13.118183) | |
| KGS03-G8 | | (78.783601, 13.121285) | (78.783601, 13.121285) | (78.783722, 13.120444) | (78.783722, 13.120444) | (78.783910, 13.119050) | (78.783910, 13.119050) | (78.783910, 13.119050) |
| KGS04-G8 | | (78.783875, 13.125500) | (78.783875, 13.125500) | (78.783990, 13.124748) | (78.783990, 13.124748) | (78.784178, 13.123354) | | |
| KGS05-G3 | | | | | (78.805950, 12.894840) | (78.805968, 12.894472) | (78.805968, 12.894472) | |
| KGS06-G3 | | | | (78.806737, 12.899868) | (78.806737, 12.899868) | (78.806755, 12.899500) | (78.806755, 12.899500) | |
| KGS07-G3 | | | | | | | (78.807424, 12.891942) | |
| KGS08-G3 | | | | | | (78.806790, 12.895180) | (78.806790, 12.895180) | (78.806790, 12.895180) |
| KGS09-G9 | | | | (78.783626, 13.120892) | (78.783626, 13.120892) | (78.783814, 13.119498) | (78.783814, 13.119498) | (78.783814, 13.119498) |
| KGS10 | | | | (78.783626, 13.120892) | (78.783626, 13.120892) | | | |
| KGS11 | | | | (78.783626, 13.120892) | | (78.783814, 13.119498) | (78.783814, 13.119498) | (78.783814, 13.119498) |
| KGS12 | | | | (78.780694, 12.996166) | (78.780694, 12.996166) | | | |
| KGS13 | | | | (78.801120, 13.098830) | (78.801120, 13.098830) | (78.801120, 13.098830) | (78.801120, 13.098830) | |
| KGS14 | | | | | | (78.806790, 12.895250) | (78.806808, 12.894882) | (78.806808, 12.894882) |
| KGS16 | | | | | | (78.806790, 12.895120) | (78.806808, 12.894882) | (78.806808, 12.894882) |
| KGS17-G5 | | | | | | (78.801270, 12.954250) | | |
| KGS18-G6 | | | | | (78.794307, 13.019443) | | (78.794307, 13.019443) | |
| KGS19-G7 | | | | | | (78.789350, 13.069880) | | |
| KGS20-G1 | | | | | | | (78.840030, 12.693890) | (78.840030, 12.693890) |
| KNG1-G6 | (78.802168, 13.009199) | | | | | | | |
| KNG2-G2 | (78.833135, 12.695976) | | | | | | | |
| KNG3-G6 | (78.792664, 13.013079) | | | | | | | |
| KNG4-G5 | (78.802964, 12.958071) | | | | | | | |
| KNG5-G7 | (78.792633, 13.057375) | | | | | | | |

Each column represents a time of field maintenance, where the station has been maintained and its position measured. Only stations associated to a group (G1 to G9) are used in this study. When no coordinates are shown, this means that the station was not deployed during this time period.

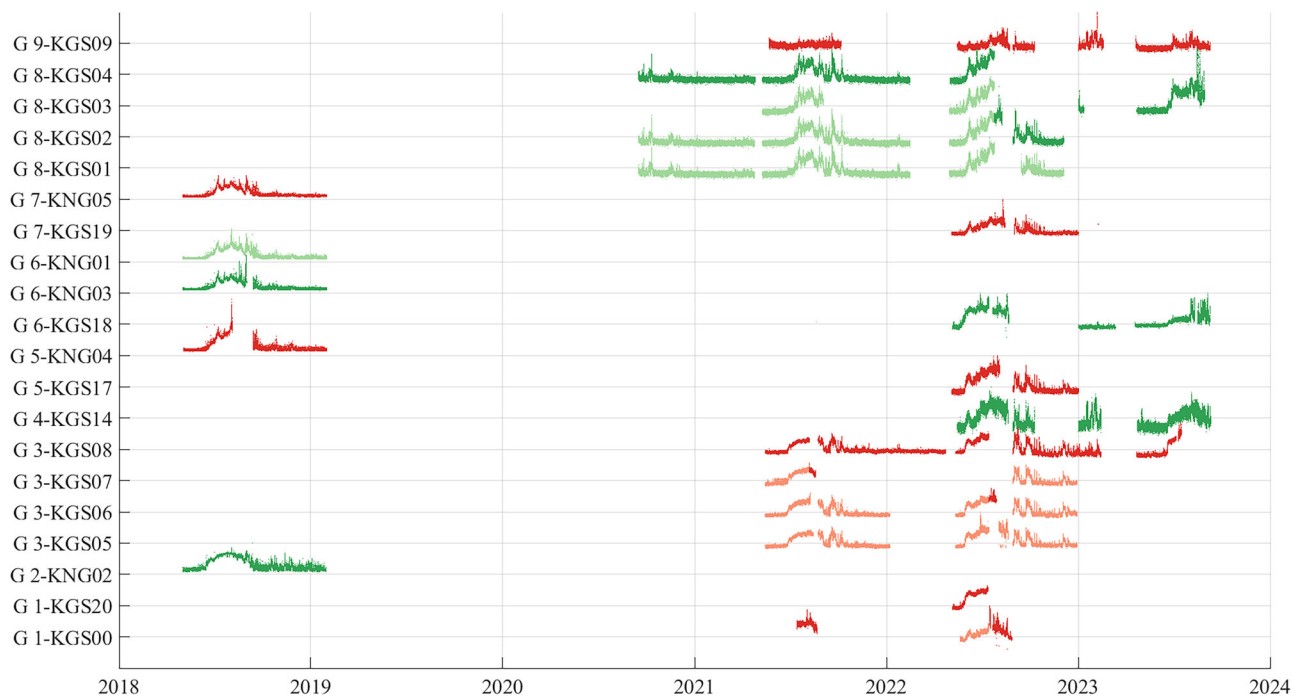

**Fig. 8 | Time series of seismic power averaged with the [5-10] Hz range.** Dark colors (dark green and dark red) show selected data within each group. The green and red colors are only used to differentiate the different groups. The group number (GX) is shown next to the station name (KGSXX) in the left axis.

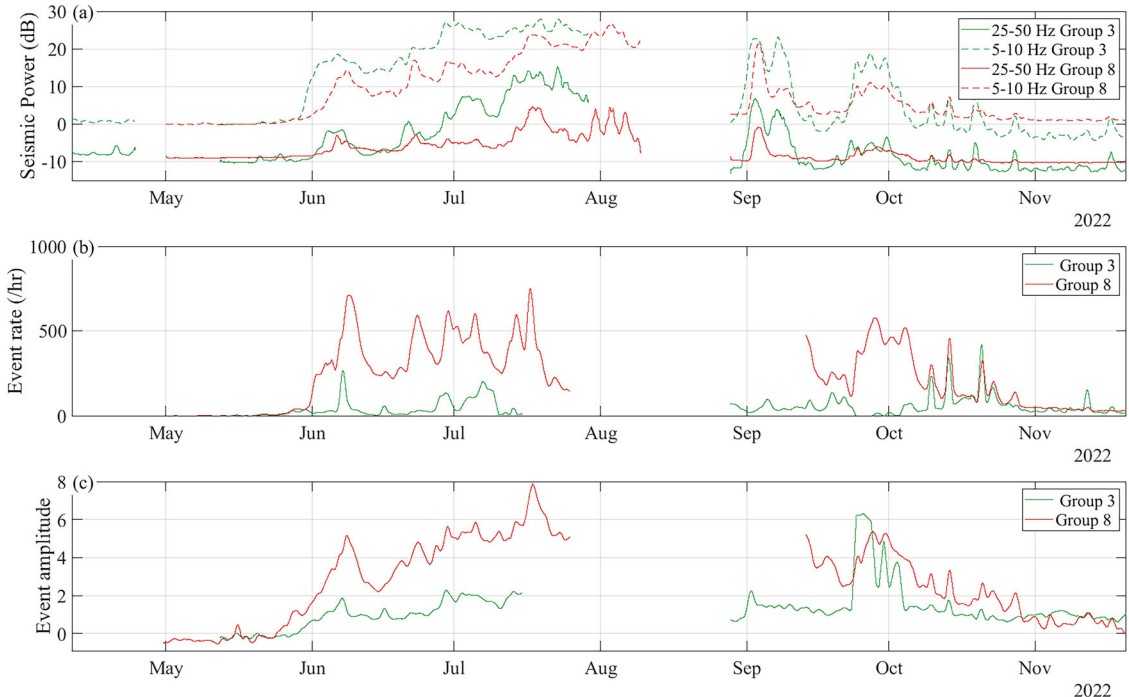

**Fig. 9 | Evaluation of the sensitivity of STA/LTA to background noise.** Seasonal evolution for group 3 (green) and 8 (red) of (**a**) background seismic noise within [5–10] Hz (dashed) and [25–20] Hz (plain); (**b**) event rate; and (**c**) event amplitude. A one day moving average is applied.

The hydraulic radius $R$ is defined as the ratio of the cross-sectional area of the channel flow $A$ to its wet perimeter $W$:

$$R = \frac{A}{W}. \tag{14}$$

This parameter scales with flow depth for open-channel flow. The hydraulic pressure gradient $S$ is a function of both the water pressure rate of change in the flow direction $\frac{\delta p}{\delta x}$ and the bed slope $\theta$:

$$S = -\frac{1}{\rho_w g}\frac{\delta p}{\delta x} + \tan\theta. \tag{15}$$

For free surface flow S equals channel slope. In a case of constant channel slope and channel geometry, increasing $S$ means closed and pressurizing channel-flow.

Reference 38 then described the subglacial water discharge $Q$ as

$$Q \propto N\beta R^{8/3} S^{1/2}. \tag{16}$$

Note that the various exponents come from a physical description of the frictional forces generated by turbulence as well as from the physical description of the turbulent flow[38].

Combining Eq. (13) and (16) and neglecting changes in $\beta$ and in $N$ (see[38] and[40] for details on theses assumptions) gives

$$
\begin{aligned}
\frac{R}{R_{\text{ref}}} &= \frac{P}{P_{\text{ref}}}^{-9/82} \frac{Q}{Q_{\text{ref}}}^{21/41}, \\
\frac{S}{S_{\text{ref}}} &= \frac{P}{P_{\text{ref}}}^{24/41} \frac{Q}{Q_{\text{ref}}}^{-30/41},
\end{aligned}
\tag{17}
$$

where the subset ref stands for a reference state, which has to be defined over the same time period for both $Q$ and $P$, but not necessarily for $R$ and $S$. Here we choose reference values averaged on June $17^{th}$ 2018 between 00:00 and 18:00 UTC at location 2. Since we do not have direct validation for $S_{\text{ref}}$ and $R_{\text{ref}}$, we present here the relative changes in the hydraulic radius and relative pressure radius and refer to $\frac{R}{R_{\text{ref}}}$ and $\frac{S}{S_{\text{ref}}}$ as $R$ and $S$ in the main text for simplicity. Note that during our entire study both $\frac{R}{R_{\text{ref}}}$ and $\frac{S}{S_{\text{ref}}}$ remain positive.

Equation (17) indicates that minor changes in $P$, i.e. in water flow turbulence, result in substantial changes in the hydraulic gradient $S$[38]. This framework was developed for conduits that form within the ice, sediments and/or over bedrock. Variations of 3 to 10 % in bed roughness relative to the water flow depth (e.g., typical for setups akin to ours;[86] could lead to uncertainties ranging from 10% to 25 %, and from 0% to 5% in estimating $S$ and $R$, respectively (for more details, please refer to the supplementary material of[38]). Changes in the number of conduits, even within one order of magnitude (a range that surpasses expectations in our setup, as reported by[87]) do not significantly impact the estimation of $S$ and $R$ (for more details, please refer to the supplementary material of ref. 40).

As shown by ref. 39, seismic noise within the selected frequency (i.e., 5–10 Hz) is dominated by turbulent water flow. The authors show that such flow happens in the distributed/inefficient drainage system, and that when a localized/efficient drainage system develops, the seismic noise generated by the turbulent water flow within such a system tends to dominates the signal. In our study, we do not make any assumptions on the geometry of the subglacial drainage system, but rather investigate the evolution of the capacity and pressure conditions through parameters than can be seen as bulk parameters representing the conditions in the subglacial drainage system averaged over one to two ice thickness, i.e., 300 to 600 m[40].

Given the selected frequency (i.e., 5–10 Hz) and the typical surface wave velocity in the ice (i.e, 1500 m s$^{-1}$;[88]), we investigate typical wavelength of c. 300 m. Because of the sensitivity of surface elastic waves being maximum within one to two wavelengths from the source[89], our seismic investigation is sensitive to turbulent water flow induced seismic noise within an area of c. 1 km$^2$ around each seismometer.

### Sensitivity of the hydraulic properties calculation to the seismometer location

Complementary to our investigation of icequakes, we have characterized the hydraulic conditions in terms of $R$ and $S$ for a given subglacial discharge $Q$ (Section "Methods";[29]), from $P$, the seismic power. Our instrumentation with co-located seismometers at the glacier surface and near its base allows us to discuss the potential combined contributions of supra-, en- and sub-glacial turbulent water flow to $P$, which has not yet been done in any glacier.

During the onset and termination of the melt seasons, the hydraulic efficiency and pressure gradient inferred from borehole data (locations 4, 9) are similar to those inferred from surface data (Fig. 3h, l, n, r). This result supports that at this time $P$ was mostly sensitive to an inefficient drainage system that enhanced water pressurization, therefore likely located subglacially. During the peak melt season, the hydraulic efficiency inferred from borehole seismometers (locations 4, 9) can be up to twice as high compared to that inferred from the surface seismometers (Fig. 3j, p), while the hydraulic gradient is half as much. This result suggests that at this time $P$ was mostly sensitive to an efficient drainage system that conveyed low-turbulent water flow near the base of the glacier, while being also influenced by a more turbulent water flow close to the surface. We interpret the origin of such a turbulent water flow at the surface to be related to the development of step-pool sequences and large bends in the supra- and en-glacial systems, as supported by the in-situ pressure observations conducted on a neighboring glacier by ref. 90. Our findings therefore reinforce the suitability of using $P$ measured both at the surface and within boreholes to investigate subglacial hydraulic properties, although surface measurements could tend to underestimate the decrease of subglacial water pressure and the increase in the subglacial drainage efficiency. In our case, this effect leads to an overestimation of the variations in $S$ ranging within 9−16% and an underestimation of $R$ ranging within 23−40% when inferred from the surface (Fig. 10). Given that changes in $R$ from the beginning of the melt season to its peak reach up to more than 2000%, we consider the analysis of $R$ and $S$ inferred from surface sensors as relevant.

Additionally, in our study, we assume that the partitioning between surface runoff $Q_{surf}$ and subglacial runoff $Q_{sub}$ is constant over the melt season. It is most likely that the ratio $\frac{Q_{sub}}{Q_{surf}}$ evolves during the season, with less water being conveyed to the bed at the beginning of the melt season than at its peak. In such a case, we would overestimate $Q_{sub}$ at the beginning of the melt season and underestimate it at the peak of the melt season. Due to eq. (17), this would imply that we would underestimate $S$ and overestimate $R$ at the beginning of the melt season, and overestimate $S$ and underestimate $R$ at the peak of the melt season. Such a change would not weaken our interpretation; instead, it would further highlight the transition from an inefficient to an efficient drainage system.

### Changes in glacier geometry

For the period 2000-2014, we use the satellite-derived surface elevation changes derived by ref. 30 for the periods 2000–2004, 2005–2009, and 2010–2014. On stable terrain, the elevation change maps have a median offset to the reference digital elevation model (DEM) of c. 0.5 m (Table S3 in ref. 30) and a normalized median absolute deviation between 0.98 and 1.42 m (Fig. S6 in ref. 30).

For the period 2014−2023, we co-register digital elevation models (DEMs) from the ArcticDEM project[91] to a well-georeferenced DEM based on aerial imagery from 2009[92]. We use all 340 ArcticDEM strips available for our study area because of the incomplete coverage of individual strips, caused by cloud cover. We co-register each strip on stable terrain (excluding all glaciers, fluvial surfaces and moraines;[93]) in two steps; first, Iterative Closest Point registration[94] and second, aspect-derived bias reduction[95]. On stable terrain, the coregistered strips maps have a Normalized Median Absolute Deviation between 1.08 m and 1.39 m.

We first reconstruct median yearly DEMs for the period 2014-2023 by blending all maps (c. 30 for each year) using an element-wise median reduction. Then, we reconstruct DEMs for the years 2000, 2005, and 2010 by subtracting the elevation changes of ref. 30 to the previously derived DEM of 2014.

For each DEM, ice thickness is then derived by subtracting basal from surface elevations using the basal topography map of ref. 74 obtained from radio-echo sounding measurements having an overall uncertainty in subglacial elevation of c. 11 m.

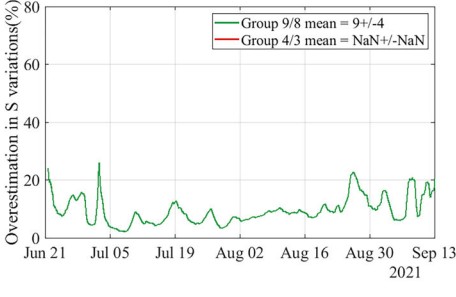
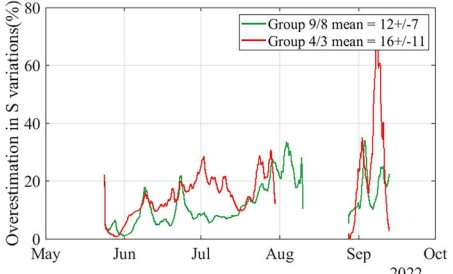
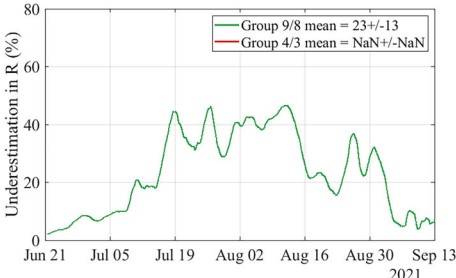
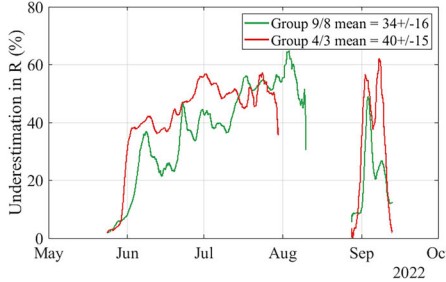

**Fig. 10 | Evaluation of the sensitivity of *R* and *S* to the seismometer location in the ice column.** (Upper panels) Comparison between the variations of *S* inferred from the borehole seismometers (location 9–green line and location 4–red line) and the variations of *S* inferred from the surface seismometers (locations 8 and 3) for the years 2021 (left panel) and 2022 (right panel). (Lower panels) Comparison between *R* inferred from the borehole seismometers (location 9–green line and location 4–red line) and *R* inferred from the surface seismometers (locations 8 and 3) for the years 2021 (left panel) and 2022 (right panel). Note that for the year 2021, only data of locations 8 and 9 are available. Note the same range in all *y*-axis.

## Data availability

The modeled runoff, glacier surface changes, glacier bed elevation, seismic power and icequake catalog have been deposited in the Zenodo database and can be accessed at https://doi.org/10.5281/zenodo.10121799[96]. Raw seismic data are registered with FDSN code N9 and will be publicly available from December 2025 and can be accessed at: https://www.fdsn.org/networks/detail/N9/. Samples of sonified seismic data from Kongsvegen can be accessed at: https://soundcloud.com/ugonanni/breath-of-arctic-glaciers.

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

## Acknowledgements

U.N., C.B., P.L., A.K. and T.S. have received support from the Research Council of Norway through the projects MAMMAMIA (grant no. 301837) and SLIDE (no. 337228). J.K. has received support from the Research Council of Norway through the projects MAMMAMIA (grant no. 301837). C.B. and F.R. have received support from the Faculty of Mathematics and Natural Sciences at the University of Oslo through the strategic research initiative Earth-Flows. T.S., O.G. and F.R. acknowledge support from the project FricFrac funded by the Center for Advanced Study at the Norwegian Academy of Science and Letters during academic year 2023-2024. U.N. acknowledge support from Circle U. 2023 seed-funding scheme. H.Å. has been supported by the project JOSTICE from the Research Council of Norway (grant no. 302458) and ERC-2022-ADG grant agreement No 01096057 GLACMASS from the European Research Council. The authors acknowledge Adrien Gilbert, Andreas Kääb, Jack Kohler, Valerie Maupin and Thomas Schellenberger for fruitful discussions. The authors acknowledge the Sverdrup Station in Ny Ålesund, and Kingsbay AS for field support and logistics. We are grateful to the Governor of Svalbard for permitting fieldwork at Kongsvegen. This study is part of the MAMMAMIA project (https://www.mn.uio.no/geo/english/research/projects/mammamia/}{Multi-scAle-Multi-Method Analysis of Mechanisms causing Ice Acceleration) led by T.S.

## Author contributions

U.N. designed and led the study. U.N. and C.B. conducted the study. U.N., C.B. and A.K. processed the seismic data. U.N. and L.S. processed the runoff data. U.N. and E.M. processed the elevation data. P.L. and J.K. processed the GNSS data. J.K. collected the long-term velocity data. U.N. and O.G. conducted the stress balance analysis. U.N. and C.B. wrote the first draft and H.Å., F.R., A.K., E.M., O.G. and T.S. contributed to refinements of the interpretations and editing of the manuscript. U.N., C.B., T.S., J. H. and A.K. participated in the field deployment of the seismic stations. J.K., P.L., T.S. and J.H. participated in the field deployment and maintenance of the GNSS stations.

## Competing interests

The authors declare no competing interests.
