## [Transparent Peer Review file · Nature Communications]

Observed positive feedback between surface ablation and crevasse formation drives glacier acceleration and potential surge

Corresponding Author: Dr Ugo nanni

Version 1:

Reviewer comments:

Reviewer #1

(Remarks to the Author)

Review of "Observed weakening..."

1. What are the noteworthy results?

This team of authors has integrated multiple data sets to suggest that a positive feedback exists between localized weakening of the bed of the glacier due to seasonal meltwater with local glacier acceleration that creates an extensional environment triggering crevasses that then allows more meltwater to reach the base of the glacier. While this can be expected in a theoretical sense, to my knowledge no one has documented it in action in this particular way.

I really appreciate that they had borehole and surface seismometers. This combination is not often used, and it adds a richness to the data set.

2. Will the work be of significance to the field and related fields?

The authors have not properly placed their work in the framework of existing knowledge and existing literature. Therefore it is hard for me to judge how significant this work will be. While there has been a lot of work in the areas of hydromechanical coupling of the glacier to its bed and of transport of water from the surface to the bed through fracturing – in Antarctica (ice streams, sticky spots, ice shelf hydrofracture, etc), in Greenland (seasonal acceleration, moulins, hydrofracture lake drainages), in Canada (i.e decades of work at Trapridge Glacier), Alaska (for example, seismicity was used to detect basal processes at Black Rapids in the 90s), yet they don't seem to cite much of this work and explain how their work fits into this existing knowledge framework. There is also rich literature in the Alps, which they do cite some of, but the introduction paragraph that is suppose to provide this framework for presenting their new ideas is written with a very broad brush that lacks specifics that show how significant this work might be.

I do think their specific feedback mechanism is intriguing, and I think the bulk of the paper is a valuable contribution, I'm not sure Nat Comm is the right place for it given the current framing of the paper and a few major weaknesses.

In particular, where this paper is weak is showing the significance of this positive feedback for glaciers/ice sheets more broadly. The authors tell us it is important, but they don't show it with evidence from observations or modeling. Specifically, they do not assess the combination of factors that would trigger this feedback elsewhere (the factors I would consider are bed slope, ice thickness, longitudinal stress/strain regime, surface mass balance, bed properties). As I mentioned above, there are theoretical reasons to believe this feedback exists, but the authors do not explore sufficiently how their observations relate to the theory. Specifically, they use a shallow ice approximation model – which ignores both longitudinal stresses and mass conservation along a flow line – yet then they conclude that longitudinal stresses are an important factor for generating the surface crevassing that leads to the positive feedback.

3. How does it compare to the established literature? If the work is not original, please provide relevant references.

First, as I mentioned above, the authors have not done a thorough job positioning their work in the literature (there are too many relevant references for me to provide here).

Second, I also often struggled to understand why they chose the citations they did. Citations may be selected because they are the first to offer a particular finding, perhaps they are an insightful review, or they have added a new novel idea to ongoing work in the community. I found it difficult to understand the intention behind certain papers being cited rather than others. Newer papers are adding a lot to the conversation every day, so they should be cited with a statement as to what new ideas they contributed - but we should also respect the knowledge contributed in the past and be specific about which paper contributed which original idea and which contributed new knowledge and how it built on other people's work.

Third, despite that I don't know who the authors are, I see many citations that appear to be self-citations – and similar to my statement above, it isn't always clear why those papers are cited over others because the sentence doesn't clarify what the specific paper contributed.

4. Does the work support the conclusions and claims, or is additional evidence needed?

I appreciated the creativity the authors used to integrate and visualize many types of data through time and space to tell the story – it is not easy. I think that the data does in general support their claims for the behavior of this glacier, but it could be told in a more straightforward way. Bringing the last figure to the beginning so that the readers know where the authors are going with their data analysis would be much easier to digest the detailed analysis.

Additional evidence (theory, models, citations to work on other glaciers) is needed to explain how this might apply to other sites. The modeling and theoretical aspects are the weakest aspect of the paper. For a journal like Nat Comm, in particular, I would expect a more thorough assessment of how their study on one glacier is relevant to other glaciers.

5. Are there any flaws in the data analysis, interpretation and conclusions? - Do these prohibit publication or require revision? - Is the methodology sound? Does the work meet the expected standards in your field?

I have three main issues with the analysis.

First, I am not convinced that the shallow ice approximation modeling shows much useful. It would not be too hard to create a flowline model that would be more rigorous. But if they do want to continue with the SIA, they really need to justify their assumptions with the data and at least back-of-the-envelope analysis (can they ignore longitudinal stresses? Does the SIA get them close enough to continuity of ice flow along the flow line?).

Second, much of the analysis relies on the reader being fully aware of (and reading) Gimbert 2016. This is fine, but to make this paper readable for those who haven't read Gimbert 2016 in detail, the authors should review key elements and equations from this paper that their analysis relies on.

Third, I do think care needs to be put on ensuring that the variability of background noise isn't affecting the identification and rates of icequakes. If the icequakes are well above the environmental background noise, this may not be an issue. But if some icequakes may be masked by background noise during some time periods and not others this can bias the comparison of ice quake rates. It seems like they used a threshold (e.g. Walters 2008), but don't explicitly say how they chose this threshold compared to the background noise variability. For detecting tiny icequakes – this can be an issue – maybe they considered a variable threshold... sort of like Carmichael 2015 (this paper isn't cited), or more sophisticated method like Carr et al 2021? Also sources of environmental noise other than the subglacial water is not discussed – such as water flowing into moulins (e.g. Rössli 2017), or surface melt/rock fall, wind on equipment, etc.

Is there enough detail provided in the methods for the work to be reproduced?

Yes, as long as the data are made available.

Reviewer #2

(Remarks to the Author)

Review of “Observed weakening of glacier ice-bed interface caused by climatic and hydro-mechanical feedbacks: towards glacier-wide acceleration?”

Summary of the manuscript

This paper uses an interesting set of observations from an alpine glacier to make inferences about the basal hydrology and its link to passive seismicity and glacier dynamics from both basal and surface seismic events. The glacier has seemingly two distinct regions 1) a lower flatter zone whose base lays below sea level and 2) an upper portion of the glacier that is steeper whose base lays above sea level. The lower portion of the glacier maintains relatively constant basal hydrologic conditions throughout the melt season while the upper portion is more dynamic in both its surface velocities and its seismic attributes, which are linked to its basal hydrologic changes. These areas of more dynamic change are hypothesized to be linked to extension in flow creating more surface crevasses which allow for greater volumes of water infiltrating to the bed that alters coupling and ice bed sliding. They term this response “hydro-mechanical feedbacks”. They use a simple estimate of basal shear stress to infer areas of increased drag and sliding speed and relate that back to their various observations.

Assessment of the manuscript

This paper is hard to follow. This in part stems from the large amount of data that are proxies for different aspects of glacier dynamics (basal shear stress, basal coupling, basal water flow, meltwater input, different regions of glacier behaving in different manners at different times of the year) and the somewhat complex explanation for linking the different aspects of the study. I found it difficult to keep all the different aspects of the paper in my head at the same time so that I could follow the explanations. After reading the paper several times it still is not clear what exactly is happening. This is complicated by the loose way glacier mechanics are applied to the observations. The complexity of the data obviously stems from the richness of the data set and the complexity of the analysis stems from the authors attempts to link the multi-component data together, however, the basal mechanics used to link the different aspects of the study is qualitative in nature. For example most of the dynamic aspects assessed within the paper are in response to transient changes throughout the season and how those changes vary from location to location on the glacier but there is essentially no mention or attempt to fit the mechanics into a quantitatively transient mechanical framework (e.g., Minchew and Meyer (2020), Tsai (2021), Zoet et al., (2021), Thøgersen et al., (2019)). The dataset is large and complex but interesting. My main concern with the paper is that the mechanical framework used to place the complex dataset into glaciological context is qualitative and thus hard to really appreciate or assess the value of the paper.

Major concerns

The observed icequakes magnitudes and their occurrence rate are not quantitatively linked to basal stress dynamics or change in stress. For example a first order question is: Do more ice quakes mean the bed is stronger, weaker, transitions from strong to weak or transitioning from weak to strong. And what is the sliding mechanism that causes this response and has it been demonstrated numerically, theoretically, experimentally, or observationally? This relationship seems to vary from place to place and time to time in the interpretation of the dataset throughout the manuscript.

There is not a strong link between the observations and basal mechanics e.g. what is “weakening” mean throughout the paper including in the title? Is this rate weakening (dealing with seismicity) or a weakening in the coupling (dealing with basal shear stress) and how specifically is this proposed to happen and how are the two linked?

Most of the paper is dealing with transient changes in slip and basal water pressure but the mechanics here don't take into account the hysteresis and lag that occurs in these quickly changing systems as the transient basal processes adjust over hours days and months. This transient nature obviously complicates the applicability of standard steady state glacier mechanics but in a system like the one monitored here the transients appear to dominate, at least for the upice portion of the glacier.

Since so much of the processes are related to the basal mechanics it is useful as a starting point to frame them as either hard or soft bed mechanics. These two systems behave differently in response to changes in meltwater in both their hydrologic handling in change in the water flux but also in their basal drag response to the changes in water pressure. I think it is necessary to place the observations in context of quantitative basal mechanics to make this paper more understandable but first it will be necessary to decide if the frameworks within soft or hard bed mechanics are more appropriate for this glacier.

The hydromechanical feedback mechanism evoked here is actually quite a standard process evoked in glacial geomorphology when referencing the formation of overdeepenings. It has not made its way prominently into glaciology but it's well established in that field. Also from the profile of this glacier bed there is a large overdeepening and so I would expect the hydromechanical feedback observed here to be the norm rather than something new.

The description of hydraulic gradient is confusing throughout the text. Its commonly referred to as increasing or decreasing. What is meant here? Is the gradient always negative but the magnitude increasing and decreasing or is it actually increasing at times and possibly changing signs from positive to negative. Typically one would assume the gradient is always negative and it is the magnitude of the gradient that is changing and not the sign but with the large overdeepening either case is possible here and thus the terminology is confusing.

From what I can tell the events do not appear to actually be located but rather they are binned as surface or bed based on which seismometers they show up on. This seems problematic as many events may be in the ice and not at the surface or the bed and locating the events would provide much greater confidence in the interpretations of the event counts changing throughout the seasons.

Is the different responses associated with the lower and upper portions resulting from the basal slope of the bed or is it because the lower portion is discharging into the ocean. This ocean boundary condition is much different than a land terminating glacier as there is a large hydrostatic pressure at the boundary that will reduce the hydrologic gradient significantly and cause it to behave differently than a land terminating glacier. These impacts need to be incorporated and considered.

The estimate for shear stress and relation to sliding is from a simple force balance (great) but with a sliding law that does not account for basal water pressure in any way. This seems problematic as its use here means that water pressure has no effect on basal dynamics which is directly in contrast to the main point of the study.

Minor comments

Figure 3 the panel labels for the upper row cover some of the text

Reviewer #3

(Remarks to the Author)

Review: Observed weakening of glacier ice-bed interface caused by climatic and hydro-mechanical feedbacks: towards glacier-wide acceleration?

Overview:

The research presents a comprehensive and succinct summary of long term geophysical, satellite and quantitative hydro-mechanical dynamics to give an insight into the mechanisms behind glacier acceleration at Kongsvegene glacier in Svalbard. The results are well presented but would benefit from some rewording of content to ensure the results, interpretation and conclusions are clearly related.

I therefore recommend this paper for publication, following the completion of some suggested changes below;

1. I feel the abstract does not really represent what the paper is showing. There is no mention of the glacier name on which the study is carried out. Further more, the abstract suggests that 'crevasse opening promoting melt water supply to the glacier base' is the main finding of the paper, which when reading the results is not clear. The abstract feels rather disconnected from the paper and it's findings.
2. Following on from this, I found the title of the paper to not fully represent what the results were showing. If the key finding is this positive feedback loop, that should be made clear in the abstract and introduction, but at the moment it is a little confusing what the main finding from the great dataset is.
3. Line 68: Please explain what a quiescent phase is.
4. Line 71: Please include what assumptions are made by stacking records from < 1 km spacing. Additionally, please can you clarify if the seismometers were re deployed each summer at the same GPS locations, or if they were deployed and moved with the glacier?
5. Line 79: I found overall that there was a lack of defining what the hydraulic terminology means in terms of the glacier. The phrases of 'subglacial hydraulic properties/pressure/radius' etc. are all used, but it is difficult to know what the author is meaning by these terminology. A description when the terms are first introduced would help the later results to be far more interpretable.
6. Figure 1: Please add lat/lon or a scale to the small figure of panel C, I am not use to studying Arctic glaciers, but found myself confused as to which map was the Arctic and which was within Svalbard. The small right figure could also benefit from being the same size as the map on the left.
7. Figure 2: Please can you define somewhere what the initiation, rise, peak, decline and termination are. Furthermore, I am not understanding why all the seismometers are not shown in all of the figures? Were some only deployed after a specific year? If this is the case, please comment on how that could have obscured your findings. Relating back to point 4. If the seismometers moved, please somehow show how the locations have evolved over the years of deployment.
8. Line 137: "indicating that icequake detection is little if at all affected by the background seismic noise level" – I do not follow how the observation in this paragraph leads to this conclusion. Please clarify in the text.
9. Section 2.3: This section would again benefit from the definitions of the hydraulic effects you're referring to. Without this, it is difficult to understand the significance of what you're observing. The definition's given in the first paragraph of section 4.5. would be useful to include earlier in the paper.
10. Line 229: The language used makes the findings feel very speculative e.g. 'likely acting'

Minor comments:

Line 332: Needs re wording for clarity e.g. "glaciers and ice caps cover 57%"

Line 326: "large area fraction" – I do not understand what is meant by this.

Line 333: space needed between 150m

Line 336: "Following the a previous surge" – remove 'a'

Line 365: "When averaging over annual period, this corresponds to April to April" – is this a typo?

I want to commend the authors for a detailed, long term, multi disciplinary study of Kongsvegene glacier, with a well thought out methodology and interesting results. I look forward to seeing the paper published,
Emma Pearce

Version 2:

Reviewer comments:

Reviewer #2

(Remarks to the Author)

This version of the manuscript denotes a significant improvement over the previous version in both the writing, organization and the analysis of the data. The writing improvements make the findings much easier to follow and the inclusion of an

analysis that has a sliding response (eqn 3) that depends on N (and thus P_w) greatly strengthens the arguments. Most of my previous comments have been addressed sufficiently. However, I still have two major concerns and several moderate concerns with the paper.

Major concerns

One major concern is that the units reported in the paper for slip and ice deformation are wrong. For example, in equation A3 (and elsewhere) the variable A_g has units of $\text{Pa}^{-2} \text{s}^{-1}$ while n was chosen to be 3 (the units should be $\text{Pa}^{-3} \text{s}^{-1}$) so all the values in the paper where stress were estimated are incorrect. It is unclear what effect this will have on the results of the paper but seemingly it should be substantial.

Second is that both equations 2 and 3 are for hard bedded glaciers while the recent work by Bouchayer et al., (2024) has shown that "Kongsvegen rests on a 5-60 m thick sediment base", thus the use of the hard bed equations (particularly eq 3) is not appropriately suited for Kongsvegen. Equation 2 is sufficiently vague with respect to physical processes that it is just generally incorrect for making predictions but is likely OK the way it is used here (where it is just inferred as some sort of process agnostic lumped friction variable). Schoof's slip law (equation 3) on the other hand is not process agnostic and instead is a guessed extension of his rate weakening solution, however a slip law of the equation 3 form was subsequently shown to be correct for hard beds when it was actually solved by Helanow et al., (2021). However, both Helanow and Schoof are only appropriate for hard beds. Somewhat conveniently Zoet and Iverson et al., (2020) proposed a regularized coulomb law with the same general form as equation 3 for till beds (like those at Kongsvegen) so I suspect much of the analysis here is generally OK but not for the mechanical reasons implied through the use of the Schoof/Gagliardini law. To make the physical processes proposed align with the observation of a soft bed it is necessary to use a soft bed slip law that is either pure Coulomb (e.g., Tulaczyk, 2000) or a regularized Coulomb like Zoet and Iverson, (2020) and Hansen et al, (2024) (which have functionally the same form as equation 3).

Furthermore, Helanow showed that C for hard beds should be between 0.1-0.2 while the value used here is 0.5. However, 0.5 is actually close to the C value you would expect for a till bed, so it seems like the modeling here, without necessarily intending to, more closely aligns with the till bed slip law of Zoet and Iverson, (2020) than it does with the hard bed mechanics of Schoof/Helanow. So, either the paper needs to make a stronger justification for why a hard bedded sliding law can be used for a till bed or equation 3 should be replaced with a soft bedded sliding law.

Moderate concerns

Nanni et al 2021 showed that seismic noise from turbulent water flow can come from linked cavity (distributed) systems and not just channels (though it does not localize to the same degree as it does in the channel) so how did the authors determine the noise they are measuring and using to estimate these hydrological properties are actually from a conduit and not a distributed system?

Minor points

Line 35. "This uncertainty is particularly due to the erratic nature of dynamic instabilities" reads awkwardly

Line 141-142. The main thing that is being evoked in affecting variable A_s within the Weertman sliding law is the water pressure so why not state that here. Note Weertman has no real physical meaning at this point as he did not consider the role of water pressure or soft beds.

Line 304-305. Stevens et al., (2024) saw this same sort of response at a glacier in Canada and their findings are pretty closely aligned with your interpretations

Fig A1. add x label (years)

Fig A3. again Pa^{-2} which does not make sense for a choice of $n=3$

Line 541-544. It would be clearer to just explicitly state the events were not located

Line 574-575. State how abundant (what % of the overall time) the times are with negative correlations

Line 586. cannot -> can no

Reviewer #4

(Remarks to the Author)
Comments attached.

Reviewer #5

(Remarks to the Author)
Please see attached referee report.

Version 3:

Reviewer comments:

Reviewer #2

(Remarks to the Author)

The authors have sufficiently addressed the comments I made in my previous statement with the new version of the text. I have noted a few small edits below

Other comments---

Line 56 therefore key for our improving our ->> therefore key for improving our

Line 174 : "Here, we assume a uniform distribution of $C = 0.3$ " but on line 499 it says $C=0.5$. update these to be the same

484: extra space after τ_l

In a few instances some of the citations have first initials that should be removed

Reviewer #4

(Remarks to the Author)

Comments attached.

Reviewer #5

(Remarks to the Author)

This paper compiles a large amount of field observations as well as inferences from force balance and a sliding law. As before, I still do not follow the logic of the paper. The authors' response was helpful in understanding their argument. however, those insights do not appear in the main text from what I can tell. If the paper is going forward, I suggest a clearer statement about the logic flow and a description of the issues that arise in the logical.

**Appendix B Reviewer #1**

**B1 General comments**

1. What are the noteworthy results?

This team of authors has integrated multiple data sets to suggest that a positive
feedback exists between localized weakening of the bed of the glacier due to seasonal melt-
water with local glacier acceleration that creates an extensional environment triggering
crevasses that then allows more meltwater to reach the base of the glacier. While this
can be expected in a theoretical sense, to my knowledge no one has documented it in ac-
tion in this particular way.

I really appreciate that they had borehole and surface seismometers. This combi-
nation is not often used, and it adds a richness to the data set.

**AR:** We thank the reviewer for this positive and encouraging comment.

2. Will the work be of significance to the field and related fields?

The authors have not properly placed their work in the framework of existing knowl-
edge and existing literature. Therefore it is hard for me to judge how significant this work
will be. While there has been a lot of work in the areas of hydromechanical coupling of
the glacier to its bed and of transport of water from the surface to the bed through frac-
turing – in Antarctica (ice streams, sticky spots, ice shelf hydrofracture, etc), in Green-
land (seasonal acceleration, moulins, hydrofracture lake drainages), in Canada (i.e decades
of work at Trapridge Glacier), Alaska (for example, seismicity was used to detect basal
processes at Black Rapids in the 90s), yet they don't seem to cite much of this work and
explain how their work fits into this existing knowledge framework. There is also rich
literature in the Alps, which they do cite some of, but the introduction paragraph that
is suppose to provide this framework for presenting their new ideas is written with a very
broad brush that lacks specifics that show how significant this work might be.

**AR:** We have now reworked the introduction (lines 36 to 74), focusing more on the con-
tribution of surges and glacier instability to the dynamic mass balance of glaciers. Hence,
we are not detailing in the introduction the processes related to hydrofracturing, but rather
focus on the surge initiation mechanisms and the influence of external conditions. Ad-
ditionally, in the discussion (lines 380 to 411) we are now comparing the mechanism we
propose to other mechanisms previously described in Svalbard to be responsible for surge
initiation, which allow us to place our study in a wider perspective and highlight its sig-
nificance.

I do think their specific feedback mechanism is intriguing, and I think the bulk of
the paper is a valuable contribution, I'm not sure Nat Comm is the right place for it given
the current framing of the paper and a few major weaknesses.

**AR:** Considering the comments from reviewers 1 and 2, we have significantly reworked
our manuscript and addressed the weakness highlighted below. To address concerns about
the manuscript's structure and flow, we have reorganized the content. The paper now
progresses logically from long-term observations of glacier velocity and elevation changes
to stress balance analysis (new section), seismic activity analysis, and an integrated dis-
cussion of the findings. We now begin with long-term observations of velocity and ele-
vation changes (Fig 1). Following this, we present a stress balance analysis of the glacier
(Fig. 2), which evaluates long-term changes in glacier dynamics and links changes in glacier
velocity to variations in basal water pressure. To investigate the causes of these changes,
we then focus on seismic analysis (Figs.3 and 4), which illustrates the development of
crevasses and basal icequakes. Finally, we discuss our findings collectively to explain the
recent acceleration of the glacier and consider broader global implications. The discus-
sion is then summarized in Fig. 5 .

Please find below, the specific changes.

In particular, where this paper is weak is showing the significance of this positive
feedback for glaciers/ice sheets more broadly. The authors tell us it is important, but
they don't show it with evidence from observations or modeling. Specifically, they do
not assess the combination of factors that would trigger this feedback elsewhere (the fac-
tors I would consider are bed slope, ice thickness, longitudinal stress/strain regime, sur-
face mass balance, bed properties).

**AR:** We agree with the reviewer's comments, and have now changed the text accord-
ingly with a new section in the Discussion, where we discuss the global relevance of our
finding (lines 380 to 411). Particularly, we discuss the potential influence of basal ther-
mal regime, climatic mass balance and changes in glacier geometry. Additionally, as de-
tailed in the answer below, we have now conducted a physical diagnosis (line 123 to 176),
which allows us to discuss the influence of internal and external drivers on the acceler-
ation of Kongsvegen.

As I mentioned above, there are theoretical reasons to believe this feedback exists,
but the authors do not explore sufficiently how their observations relate to the theory.
Specifically, they use a shallow ice approximation model which ignores both longitudi-
nal stresses and mass conservation along a flow line – yet then they conclude that lon-
gitudinal stresses are an important factor for generating the surface crevassing that leads
to the positive feedback.

**AR:** We appreciate the reviewer's comments regarding our initial physical approach,
an agree that it might have been too simple. In response, we have revised our analysis
to provide a more detailed investigation of stress balance and basal conditions, and have
added a new section in Sect. 2.2 (line 123 to 176). We now specifically address the as-
sumptions used in estimating basal shear stress and assess the effects of both lateral and
longitudinal stresses. Initially, we apply Weertman (1957)'s sliding law to evaluate changes
in the effective sliding parameter, and then incorporate a more detailed sliding law to
explore the impact of basal water pressure. This revised approach aligns with the method-
ology of Jay-Allemand et al. (2011), which was used to model the evolution of the Var-
iegated Glacier surge. We also acknowledge the importance of longitudinal stresses in
generating surface crevassing, which contributes to the positive feedback mechanism. Our
updated analysis now explicitly considers this factor, addressing the theoretical impli-
cations and ensuring a stronger connection between our observations and the underly-
ing theory.

3. How does it compare to the established literature? If the work is not original,
please provide relevant references.

First, as I mentioned above, the authors have not done a thorough job position-
ing their work in the literature (there are too many relevant references for me to pro-
vide here).

Second, I also often struggled to understand why they chose the citations they did.
Citations may be selected because they are the first to offer a particular finding, perhaps
they are an insightful review, or they have added a new novel idea to ongoing work in
the community. I found it difficult to understand the intention behind certain papers be-
ing cited rather than others. Newer papers are adding a lot to the conversation every
167 day, so they should be cited with a statement as to what new ideas they contributed -
168 but we should also respect the knowledge contributed in the past and be specific about
169 which paper contributed which original idea and which contributed new knowledge and
170 how it built on other people's work.

**AR:** Given the new structure of the paper with major changes in the introduction and
in the discussion, we have now also changed a significant part of the references. When
citing papers, we carefully check the reason for which we selected a given reference, ei-
ther we searched for the 'original' reference or for the state of the art, depending on the
context.

Third, despite that I don't know who the authors are, I see many citations that ap-
 pear to be self-citations – and similar to my statement above, it isn't always clear why
 those papers are cited over others because the sentence doesn't clarify what the specific
 paper contributed.

**AR:** We have retained four references to studies of several coauthors Gimbert et al. (2016);
 Thøgersen et al. (2019); Nanni et al. (2020, 2021) where necessary, as our methodology
 is directly based on these studies. Additionally, we have kept the citation of Bouchayer
 et al. (2024) because it represents the most recent and comprehensive study conducted
 on the Kongsvegen's glacier. However, we have removed any citations that were not es-
 sential to the context, ensuring that each citation clearly supports the specific point be-
 ing made.

4. Does the work support the conclusions and claims, or is additional evidence needed?

I appreciated the creativity the authors used to integrate and visualize many types
 of data through time and space to tell the story – it is not easy. I think that the data
 does in general support their claims for the behavior of this glacier, but it could be told
 in a more straightforward way. Bringing the last figure to the beginning so that the read-
 ers knows where the authors are going with their data analysis would be much easier to
 digest the detailed analysis.

**AR:** We appreciate the reviewer's feedback on the clarity of our manuscript. We have
 reorganized the structure of the paper and reordered the figures to provide a clearer nar-
 rative for the reader. We now start with long-term observations of velocity and eleva-
 tion changes (Fig.1). Next, we present a stress balance analysis (Fig. 2), which exam-
 ines long-term changes in glacier dynamics and connects these to variations in basal wa-
 ter pressure. To further explore the underlying causes of these changes, we then focus
 on seismic analysis (Figs.3 and 4), highlighting the development of crevasses and basal
 icequakes. Finally, we summarize our discussion of the glacier's recent acceleration and
 its broader implications (Fig. 5). We believe this new structure provides a more straight-
 forward and consistent narrative, making it easier for the reader to follow our analysis
 and conclusions.

Additional evidence (theory, models, citations to work on other glaciers) is needed
 to explain how this might apply to other sites. The modeling and theoretical aspects are
 the weakest aspect of the paper. For a journal like Nat Comm, in particular, I would ex-
 pect a more thorough assessment of how their study on one glacier is relevant to other
 glaciers.

**AR:** We agree with the reviewer's comments, and have now changed the text accord-
 ingly with a new section in the Discussion, where we discuss the global relevance of our
 finding (lines 380 to 411).

5. Are there any flaws in the data analysis, interpretation and conclusions? - Do
 these prohibit publication or require revision?- Is the methodology sound? Does the work
 meet the expected standards in your field?

I have three main issues with the analysis.

First, I am not convinced that the shallow ice approximation modeling shows much
 useful. It would not be too hard to create a flowline model that would be more rigor-
 ous. But if they do want to continue with the SIA, they really need to justify their as-
 sumptions with the data and at least back-of-the-envelope analysis (can they ignore lon-
 gitudinal stresses? Does the SIA get them close enough to continuity of ice flow along
 the flow line?).

**AR:** As described above, we have now adopted a more detailed mechanics framework.
 Particularly, we have revised our analysis to provide a more detailed investigation of stress
 balance and basal conditions, and have added a new section in Sect. 2.2 (line 123 to 176).

Second, much of the analysis relies on the reader being fully aware of (and read-
 ing) Gimbert 2016. This is fine, but to make this paper readable for those who haven't
 read Gimbert 2016 in detail, the authors should review key elements and equations from

this paper that their analysis relies on. Also sources of environmental noise other than
the subglacial water is not discussed – such as water flowing into moulines (e.g. Rössli
2017), or surface melt/rock fall, wind on equipment, etc.

**AR:** We have addressed this concern by adding more detailed explanations of the method-
ology, both in the main text (lines 187 to 202) and in the Methods section (lines 603 to
633), to ensure that readers who are not familiar with Gimbert (2016) can fully under-
stand the analysis.

Third, I do think care needs to be put on ensuring that the variability of background
noise isn't affecting the identification and rates of icequakes. If the icequakes are well
above the environmental background noise, this may not be an issue. But if some ice-
quakes may be masked by background noise during some time periods and not others
this can bias the comparison of ice quake rates. It seems like they used a threshold (e.g.
Walters 2008), but don't explicitly say how they chose this threshold compared to the
background noise variability. For detecting tiny icequakes – this can be an issue – maybe
they considered a variable threshold... sort of like Carmichael 2015 (this paper isn't cited),
or more sophisticated method like Carr et al 2021?

**AR:** We have now clarified this point in the main text (line 208) as:

We opted for a constant threshold since we did not observe a significant influence
of background seismic noise on the icequake detection.

**AR:** We have also added details on the Method Section, lines 564 to 601, where we specif-
ically analyze the influence of noise on event detections.

Is there enough detail provided in the methods for the work to be reproduced? Yes,
as long as the data are made available.

**AR:** The data are made available as described in the Acknowledgment section (line 718).

Appendix C Reviewer #2

C1 General comments

Review of “ Observed weakening of glacier ice-bed interface caused by climatic and hydro-mechanical feedbacks: towards glacier-wide acceleration?”

Summary of the manuscript

This paper uses an interesting set of observations from an alpine glacier to make inferences about the basal hydrology and its link to passive seismicity and glacier dynamics from both basal and surface seismic events. The glacier has seemingly two distinct regions 1) a lower flatter zone whose base lays below sea level and 2) an upper portion of the glacier that is steeper whose base lays above sea level. The lower portion of the glacier maintains relatively constant basal hydrologic conditions throughout the melt season while the upper portion is more dynamic in both its surface velocities and its seismic attributes, which are linked to its basal hydrologic changes. These areas of more dynamic change are hypothesized to be linked to extension in flow creating more surface crevasses which allow for greater volumes of water infiltrating to the bed that alters coupling and ice bed sliding. They term this response “hydro-mechanical feedbacks”. They use a simple estimate of basal shear stress to infer areas of increased drag and sliding speed and relate that back to their various observations.

Assessment of the manuscript

This paper is hard to follow. This in part stems from the large amount of data that are proxies for different aspects of glacier dynamics (basal shear stress, basal coupling, basal water flow, meltwater input, different regions of glacier behaving in different manners at different times of the year) and the somewhat complex explanation for linking the different aspects of the study. I found it difficult to keep all the different aspects of the paper in my head at the same time so that I could follow the explanations. After reading the paper several times it still is not clear what exactly is happening. This is complicated by the loose way glacier mechanics are applied to the observations.

AR: We acknowledge that the previous version of our manuscript was difficult to follow and make here the same answer as for Reviewer 1, who also raised this issue. To improve clarity, we have reorganized the paper’s structure and reordered the figures to better guide the reader. We now begin with long-term observations of velocity and elevation changes (Fig 1). Following this, we present a stress balance analysis of the glacier (Fig. 2), which evaluates long-term changes in glacier dynamics and links changes in glacier velocity to variations in basal water pressure. To investigate the causes of these changes, we then focus on seismic analysis (Figs.3 and 4), which illustrates the development of crevasses and basal icequakes. Finally, we discuss our findings collectively to explain the recent acceleration of the glacier and consider broader global implications. The discussion is then summarized in Fig. 5 .

The complexity of the data obviously stems from the richness of the data set and the complexity of the analysis stems from the authors attempts to link the multi-component data together, however, the basal mechanics used to link the different aspects of the study is qualitative in nature. For example most of the dynamic aspects assessed within the paper are in response to transient changes throughout the season and how those changes vary from location to location on the glacier but there is essentially no mention or attempt to fit the mechanics into a quantitatively transient mechanical framework (e.g., Minchew and Meyer (2020), Tsai (2021), Zoet et al., (2021), Thøgersen et al., (2019)). The dataset is large and complex but interesting. My main concern with the paper is that the mechanical framework used to place the complex dataset into glaciological context is qualitative and thus hard to really appreciate or assess the value of the paper.

AR: We appreciate the reviewer’s feedback regarding the limitations of our previous physical approach. In response, we have significantly reworked our analysis to include a more

quantitative and detailed investigation of stress balance and basal conditions, as detailed
 in Sect 2.2 - lines 123 to 176. Specifically, we now elaborate on the assumptions made
 in estimating basal shear stress, including the effects of lateral and longitudinal stresses.
 We begin by applying Weertman (1957)'s sliding law to assess changes in the effective
 sliding parameter, and then proceed to use a more comprehensive Schoof (2005)-type slid-
 ing law to examine the influence of basal water pressure. Our approach parallels the method-
 ology of Jay-Allemand et al. (2011), which was employed to model the evolution of the
 Variegated Glacier surge.

We opted not to use a transient mechanical framework because our primary goal
 is to identify the processes driving the acceleration of Kongsvegen, which we explore fur-
 ther through our seismic analysis. We believe that the new structure of the paper bet-
 ter guides the reader through our methodology and clarifies the application of our phys-
 ical framework. Finally, we discuss the potential future behavior of Kongsvegen's glacier
 in light of the models presented by Thøgersen et al. (2019).

Major concerns The observed icequakes magnitudes and their occurrence rate are
 not quantitatively linked to basal stress dynamics or change in stress. For example a first
 order question is: Do more ice quakes mean the bed is stronger, weaker, transitions from
 strong to weak or transitioning from weak to strong. And what is the sliding mechanism
 that causes this response and has it been demonstrated numerically, theoretically, ex-
 perimentally, or observationally? This relationship seems to vary from place to place and
 time to time in the interpretation of the dataset throughout the manuscript.

**AR:** We now properly define the link between basal icequakes and changes in basal stress
 conditions in Section 3 - lines 310 to 326. Particularly, we analyze our results in the light
 of experimental and theoretical study on basal icequakes(Lipovsky & Dunham, 2016; Gräff
 et al., 2021; L. Zoet et al., 2020).

There is not a strong link between the observations and basal mechanics e.g. what
 is "weakening" mean throughout the paper including in the title? Is this rate weaken-
 ing (dealing with seismicity) or a weakening in the coupling (dealing with basal shear stress)
 and how specifically is this proposed to happen and how are the two linked?

**AR:** We have revised the manuscript to avoid using the term "weakening" in the title.
 We now use only the term "rate-weakening frictional regime" specifically when discussing
 basal icequakes (line 319).

Most of the paper is dealing with transient changes in slip and basal water pres-
 sure but the mechanics here don't take into account the hysteresis and lag that occurs
 in these quickly changing systems as the transient basal processes adjust over hours days
 and months. This transient nature obviously complicates the applicability of standard
 steady state glacier mechanics but in a system like the one monitored here the transients
 appear to dominate, at least for the upice portion of the glacier.

**AR:** We now investigate in Sect 2.2 (lines 123 to 176) a multi-year view on the glacier
 mechanics involved in the aforementioned acceleration, over such timescales the system
 can be considered to be at steady state and therefore we do not need to take into account
 potential hysteresis. When we investigate shorter time scales, we do not investigate tran-
 sient changes in slip and basal water pressure, but rather the evolution of the subglacial
 drainage system and the occurrence of icequakes.

Since so much of the processes are related to the basal mechanics it is useful as a
 starting point to frame them as either hard or soft bed mechanics. These two systems
 behave differently in response to changes in meltwater in both their hydrologic handling
 in change in the water flux but also in their basal drag response to the changes in wa-
 ter pressure. I think it is necessary to place the observations in context of quantitative
 basal mechanics to make this paper more understandable but first it will be necessary
 to decide if the frameworks within soft or hard bed mechanics are more appropriate for
 this glacier.

AR: Given the new structure of the paper, we discuss in detail in the new section Sect 2.2 (lines 123 to 176) our choice on the sliding laws (Weertam-type and Schoof-type) in the context of Kongsvegen’s bed. Additionally, we also discuss in the Discussion (lines 310 to 326) the potential influence of basal sediments on the glacier sliding behaviors.

The hydromechanical feedback mechanism evoked here is actually quite a standard process evoked in glacial geomorphology when referencing the formation of overdeepenings. It has not made its way prominently into glaciology but it’s well established in that field. Also from the profile of this glacier bed there is a large overdeepening and so I would expect the hydromechanical feedback observed here to be the norm rather than something new.

AR: The glacier profile shown in Fig.1c, indeed shows an over-deepening from 12 to 14 km; however, this over-deepening is not exaggerated due to the vertical exaggeration of 10 in our figure. When compared to neighboring glaciers (Fig. 7 in Lindbäck et al. (2018)) or literature about over-deepening (Patton et al., 2016), we can see that it is indeed a rather insignificant over-deepening.

The description of hydraulic gradient is confusing throughout the text. Its commonly referred to as increasing or decreasing. What is meant here? Is the gradient always negative but the magnitude increasing and decreasing or is it actually increasing at times and possibly changing signs from positive to negative. Typically one would assume the gradient is always negative and it is the magnitude of the gradient that is changing and not the sign but with the large overdeepening either case is possible here and thus the terminology is confusing.

AR: We now clearly define the hydraulic pressure gradient on line 491 as: The hydraulic gradient combines the gradient of subglacial water pressure and the bed slope (Methods, Eq. C1), such that an increase in the relative hydraulic gradient (S ; color-coded in Fig. 3) can be interpreted as a pressurization of the subglacial drainage system (Gimbert et al., 2016).

AR: We also precised in the Method section in Eq. C1 the definition of the hydraulic pressure gradient, and therefore the meaning of its sign:

The hydraulic pressure gradient S is a function of both the water pressure rate of change in the flow direction $\frac{\delta p}{\delta x}$ and the bed slope θ :

$$S = -\frac{1}{\rho_w g} \frac{\delta p}{\delta x} + \tan \theta. \quad (\text{C1})$$

For free surface flow S equals channel slope. In a case of constant channel slope and channel geometry, increasing S means closed and pressurizing channel-flow.

From what I can tell the events do not appear to actually be located but rather they are binned as surface or bed based on which seismometers they show up on. This seems problematic as many events may be in the ice and not at the surface or the bed and locating the events would provide much greater confidence in the interpretations of the event counts changing throughout the seasons.

AR: We now better explain this aspect in the main text from line 211:

We did not localize these icequakes; instead, we differentiated between near-surface icequake, typically attributed to surface crevassing (Walter et al., 2008) and near-bed icequakes, typically attributed to basal crevassing and stick-slip events (Gräff et al., 2021) by comparing records from co-located surface and borehole seismometers. [...] For icequake activity, we observe that the icequake rates are generally lower when inferred from the basal seismic records than from the surface records, whereas the amplitudes recorded by the basal seismometers tend to be higher, particularly during and after peak melt-season (Fig. 4, k, l, p, q). Specifically, at locations 8 and 9 and during the 2022 and 2023 seasons, the icequakes detected by the surface seismometers are about ten times more frequent (3,647,186) than those detected by the borehole seismometers (346,408). Only 20% of the borehole-detected icequakes are also detected by the surface seismometers. These differences indicate the dominant sensitivity of surface seismometers to near-surface icequakes and of borehole seismometers to near-bed icequakes, similar to observations at other glaciers (Walter et al., 2008; Gräff et al., 2021).

**AR:** We also precise in the caption of Fig. 4: Spatio-temporal evolution of icequake ac-
 tivity shown at the measurement locations.

Is the different responses associated with the lower and upper portions resulting
 from the basal slope of the bed or is it because the lower portion is discharging into the
 ocean. This ocean boundary condition is much different than a land terminating glacier
 as there is a large hydrostatic pressure at the boundary that will reduce the hydrologic
 gradient significantly and cause it to behave differently than a land terminating glacier.
 These impacts need to be incorporated and considered.

**AR:** While we acknowledge that the boundary conditions at the ocean-terminating lower
 portion of the glacier differ from those at a land-terminating glacier, these differences
 do not significantly affect our results for several reasons:

The ocean depth at the terminus is approximately 50 m, which modifies the pressure
 gradient minimally. For example, with a 50 m water column at the terminus and
 a top glacier elevation of 600 m, the pressure gradient would be $\Delta P = \frac{600-50\text{m}}{15\text{km}}$ instead
 of $\Delta P = \frac{600-0\text{m}}{15\text{km}}$ if we do not take into account the water column. Hence, this is a neg-
 ligible difference in the overall gradient.

The pressure gradient steepens towards the glacier front (see Linback, 2018, for ex-
 amples of exposed bed conditions). These localized effects are primarily relevant near
 the glacier terminus and do not influence the regions farther upstream.

Our analysis focuses on **temporal changes** in the pressure gradient at a local scale
 (within $\tilde{1}$ km of each seismic station). Consequently, our results are based on relative
 changes to a given state rather than absolute pressure gradients, making boundary con-
 dition differences irrelevant.

Additionally, our framework does not assume a specific shape for the pressure gra-
 dient along the centerline and therefore does not rely on predefined boundary conditions.

The estimate for shear stress and relation to sliding is from a simple force balance
 (great) but with a sliding law that does not account for basal water pressure in any way.
 This seems problematic as its use here means that water pressure has no effect on basal
 dynamics which is directly in contrast to the main point of the study.

**AR:** As detailed on a previous comment (line 302 of the response to reviewers) we now
 use a Schoof-type sliding law that explicitly accounts for the basal water pressure.

Minor comments Figure 3 the panel labels for the upper row cover some of the text

**AR:** This has been changed.

Appendix D Reviewer #3 - E. Pearce

D1 General comments

The research presents a comprehensive and succinct summary of long term geophysical, satellite and quantitative hydro-mechanical dynamics to give an insight into the mechanisms behind glacier acceleration at Kongsvegene glacier in Svalbard. The results are well presented but would benefit from some rewording of content to ensure the results, interpretation and conclusions are clearly related. I therefore recommend this paper for publication, following the completion of some suggested changes below;

1. I feel the abstract does not really represent what the paper is showing. There is no mention of the glacier name on which the study is carried out. Further more, the abstract suggests that ‘crevasse opening promoting melt water supply to the glacier base’ is the main finding of the paper, which when reading the results is not clear. The abstract feels rather disconnected from the paper and it’s findings.

AR: We have now changed the abstract has following this comment and following the whole reframing of our study (line 208 to 34).

2. Following on from this, I found the title of the paper to not fully represent what the results were showing. If the key finding is this positive feedback loop, that should be made clear in the abstract and introduction, but at the moment it is a little confusing what the main finding from the great dataset is.

AR: We have now changed the title.

3. Line 68: Please explain what a quiescent phase is.

AR: This has been changed to: [...] has since exhibited a slow and stable flow, with annual surface velocities lower than 5 m yr^{-1} [...]

4. Line 71: Please include what assumptions are made by stacking records from $\geq 1 \text{ km}$ spacing. Additionally, please can you clarify if the seismometers were re deployed each summer at the same GPS locations, or if they were deployed and moved with the glacier?

AR: We have now added in the Methods section the Table A1 with all the seismometers availability and position through time.

We also show in Fig.A4 the details on how we stack the records for each group, illustrated with the seismic power. We select the record with the best quality, and observe that when two records from neighboring stations are available at the same time, they are very similar. This observation shows that the grouping approach allows us to keep the best data, with a limited influence on the analysis. In Fig.A4, we show the seismic power recorded at the 20 seismic stations, each associated to a group from G1 to G9 as in Table A1.

5. Line 79: I found overall that there was a lack of defining what the hydraulic terminology means in terms of the glacier. The phrases of. ‘subglacial hydraulic properties/pressure/radius’ etc. are all used, but it is difficult to know what the author is meaning by these terminology. A description when the terms are first introduced would help the later results to be far more interpretable.

AR: We have now reworked the paragraph in Sect. 2.3 such as:

We analyzed the subglacial hydraulic properties at each seismometer group using the continuous seismic signal. Our focus was on the vertical component of the ground velocity within the frequency band 5 to 10 Hz, a range typically dominated by tremor induced by turbulent water flow (Burtin et al., 2008; Gimbert et al., 2014). We adopt the framework of Gimbert et al. (2016), based on a forward model relating relative changes in seismic power and subglacial runoff to relative changes in subglacial hydraulic radius and pressure gradient (Methods, Eq. A14). The hydraulic radius expresses the influence of wall friction for a given cross-section (Methods, Eq. A11), such that an increase in relative hydraulic radius (R ; size-coded in Fig. 3) can be interpreted as an increase in subglacial drainage capacity (Nanni et al., 2020). The hydraulic gradient combines the gra-

492 dent of subglacial water pressure and the bed slope (Methods, Eq. C1), such that an
 493 increase in the relative hydraulic gradient (S ; color-coded in Fig. 3) can be interpreted
 as a pressurization of the subglacial drainage system (Gimbert et al., 2016). Here, we
 retrieved changes in subglacial hydraulic properties relative to a reference state, defined
 as June 17th 2018 between 00:00 and 18:00 UTC at location 2.

6. Figure 1: Please add lat/lon or a scale to the small figure of panel C, I am not
 use to studying Arctic glaciers, but found myself confused as to which map was the Arc-
 tic and which was within Svalbard. The small right figure could also benefit from be-
 ing the same size as the map on the left.

**AR:** We have now modified Fig. 1 and added a scale on the map of Svalbard as well as
 recalled the coordinates of Kongsvegen glacier in the associated legend.

7. Figure 2: Please can you define somewhere what the initiation, rise, peak, de-
 cline and termination are. Furthermore, I am not understanding why all the seismome-
 ters are not shown in all of the figures? Were some only deployed after a specific year?
 If this is the case, please comment on how that could have obscured your findings. Re-
 lating back to point 4. If the seismometers moved, please somehow show how the loca-
 tions have evolved over the years of deployment.

**AR:** We have now changed the orders of the figures, so the old Figure 2 (seismic events)
 is now Fig.4, which comes after Fig. 3 where we defined the periods as follows:

[Fig. 3 caption] The periods are chosen to capture the initiation, rise, peak, decline
 and termination of each melt season.

[Fig. 4 caption] Surface velocity at locations 1 (down-glacier, salmon) and 8 (up-
 glacier, light red) with the same five selected periods (dark red) as shown in Fig.3.

**AR:** We have now added in the Methods section a Table with all the seismometers avail-
 ability as not all seismometers were deployed at the same time, and have added this pre-
 cision in the caption of Fig. 3:

[Fig. 3 caption] Because of instrument deployment and availability, not all seismome-
 ters groups cover the entire period (Table A1).

8. Line 137: “indicating that icequake detection is little If at all affected by the back-
 ground seismic noise level” – I do not follow how the observation in this paragraph leads
 to this conclusion. Please clarify in the text.

**AR:** We have now clarified the text in the main text as:

We chose here a constant threshold as we do not see a clear influence of background
 seismic noise on the icequake detection (Methods, Sect. A6).

**AR:** We have also added details on the Method Section A6, where we describe the sen-
 sitivity of our approach.

9. Section 2.3: This section would again benefit from the definitions of the hydraulic
 effects you’re referring to. Without this, it is difficult to understand the significance of
 what you’re observing. The definition’s given in the first paragraph of section 4.5. would
 be useful to include earlier in the paper.

**AR:** As detailed in comment 5, we have now substantially modified the structure of the
 paper, so the hydraulic properties are now defined when we describe our observations.

10. Line 229: The language used makes the findings feel very speculative e.g. ‘likely
 acting’

**AR:** Given the new structure of the paper, we have now removed this sentence.

I want to commend the authors for a detailed, long term, multi disciplinary study
 of Kongsvegene glacier, with a well thought out methodology and interesting results. I
 look forward to seeing the paper published, Emma Pearce

D2 Specific comments

Line 332: Needs re wording for clarity e.g. “glaciers and ice caps cover 57%”

**AR:** This has been changed to: [...] glaciers and ice caps cover 57% of its 60,000 km²
 land surface area [...]

Line 326: “large area fraction” – I do not understand what is meant by this.

**AR:** This has been changed to: [...] Due to a large fraction of the glaciated areas be-
ing located at low elevation [...]

Line 333: space needed between 150m

**AR:** Given the substantial revision of the manuscript, this specific comment is no longer
applicable.

Line 336: “Following the a previous surge” – remove ‘a’

**AR:** This has been corrected.

Line 365: “When averaging over annual period, this corresponds to April to April”
– is this a typo?

**AR:** This has been changed to: [...] this corresponds to April of a given year to April
of the following year [...]

Line 580 “Atlantification”; please explain.

**AR:** Given the substantial revision of the manuscript, this specific comment is no longer
applicable.

Reviewer #1

This version of the manuscript denotes a significant improvement over the previous version in both the writing, organization and the analysis of the data. The writing improvements make the findings much easier to follow and the inclusion of an analysis that has a sliding response (eqn 3) that depends on N (and thus P_w) greatly strengthens the arguments. Most of my previous comments have been addressed sufficiently. However, I still have two major concerns and several moderate concerns with the paper.

Major concerns

One major concern is that the units reported in the paper for slip and ice deformation are wrong. For example, in equation A3 (and elsewhere) the variable A_g has units of $\text{Pa}^{-2} \text{s}^{-1}$ while n was chosen to be 3 (the units should be $\text{Pa}^{-3} \text{s}^{-1}$) so all the values in the paper where stress were estimated are incorrect. It is unclear what effect this will have on the results of the paper but seemingly it should be substantial.

AR: We thank the reviewer for pointing out this aspect. This was a typo mistake we made in Figure A3 and in equation A3. Throughout the rest of the manuscript, the variable of A_g has units of $\text{Pa}^{-3} \text{s}^{-1}$ as in Cuffey and Paterson (2010). We have corrected this in Figure A3 and in equation A3, and this has no effect on our results.

Second is that both equations 2 and 3 are for hard bedded glaciers while the recent work by Bouchayer et al., (2024) has shown that “Kongsvegen rests on a 5-60 m thick sediment base”, thus the use of the hard bed equations (particularly eq 3) is not appropriately suited for Kongsvegen. Equation 2 is sufficiently vague with respect to physical processes that it is just generally incorrect for making predictions but is likely OK the way it is used here (where it is just inferred as some sort of process agnostic lumped friction variable).

AR: First, about the use of a Weertman type sliding law. We use in equation 2 (line 158) a Weertman type sliding law, which, as pointed out by the reviewer is appropriate since we are not making predictions, but are using this law to infer general changes in basal friction. Our goal by using this law is to reconstruct the evolution of A_s , which we treat here as a bulk parameter accounting for various potential basal processes, such as sediment deformation and basal hydrology effects (e.g. basal water pressure), and therefore we do not consider a particular bed type. s

We clarify this is the manuscript as:

Line (line 158) : We now retrieve the evolution of Kongsvegen’s basal friction conditions from the reconstructed τ_b . First, we reconstruct the evolution of A_s , a bulk basal sliding parameter; the higher A_s the more the glacier slips (Fig. 2c). This parameter and accounts for various potential basal processes, such as sediment deformation and basal hydrology effects (e.g. basal water pressure). To calculate A_s we use a simple Weertman (1957)-type sliding law, which can be considered agnostic to bed type. [...]

Although the evolution of A_s highlights a reduction in basal friction since 2014, the formulation of Weertman (1957) does not allow distinguishing the influences of substrate and those of hydraulic conditions.

Schoof’s slip law (equation 3) on the other hand is not process agnostic and instead is a guessed extension of his rate weakening solution, however a slip law of the equation 3 form was subsequently shown to be correct for hard beds when it was actually solved by Helanow et al., (2021). However, both Helanow and Schoof are only appropriate for hard beds. Somewhat conveniently Zoet and Iverson et al., (2020) proposed a regularized coulomb law with the same general form as equation 3 for till beds (like those at Kongsvegen) so I suspect much of the analysis here is generally OK but not for the mechanical reasons implied through the use of the Schoof/Gagliardini law. To make the physical processes proposed align with the observation of a soft bed it is necessary to use a

soft bed slip law that is either pure Coulomb (e.g., Tulaczyk, 2000) or a regularized Coulomb like Zoet and Iverson, (2020) and Hansen et al, (2024) (which have functionally the same form as equation 3).

Furthermore, Helanow showed that C for hard beds should be between 0.1-0.2 while the value used here is 0.5. However, 0.5 is actually close to the C value you would expect for a till bed, so it seems like the modeling here, without necessarily intending to, more closely aligns with the till bed slip law of Zoet and Iverson, (2020) than it does with the hard bed mechanics of Schoof/Helanow. So, either the paper needs to make a stronger justification for why a hard bedded sliding law can be used for a till bed or equation 3 should be replaced with a soft bedded sliding law.

AR: Second, about the use of a schoof type slip law in Equation 3, as the reviewer says, L. Zoet and Iverson (2020) has shown that such a formulation holds for also for sediment-type bed. Therefore, we keep the same formulation. We also use a value of $C = 0.3$, as suggested by Thøgersen et al. (2019) for soft-bed glacier. We have clarified in the main text as follows, from line 172:

To disentangle these effects, we adopt the approach of Jay-Allemand et al. (2011) and implement a water pressure-dependent sliding law that explicitly accounts for bed friction properties. L. Zoet and Iverson (2020) have proposed a regularized coulomb sliding law appropriate for till bed mechanics that has a similar formulation as the one proposed by Schoof (2005) and Helanow et al. (2021) for hard bed mechanics. Considering C to be a constant related to bed roughness and A a bed friction coefficient, we use the following formulation to diagnose the evolution of Kongsvegen’s basal water pressure conditions:

$$\tau_b = CN \left(\frac{U_b}{U_b + AC^n N^n} \right)^{1/n}. \quad (\text{A15})$$

Where N the effective pressure difference between the ice overburden pressure $P_i = \rho_i g H$ and the basal water pressure P_w) and the transition. Here, we assume a uniform distribution of $C = 0.3$, a value likely expected for sediment-type bed (Thøgersen et al., 2019; Helanow et al., 2021)

Moderate concerns

Nanni et al 2021 showed that seismic noise from turbulent water flow can come from linked cavity (distributed) systems and not just channels (though it does not localize to the same degree as it does in the channel) so how did the authors determine the noise they are measuring and using to estimate these hydrological properties are actually from a conduit and not a distributed system?

AR: We thank the reviewer for raising up this point. Indeed in Nanni et al., (2021), the authors show that seismic noise recorded at the surface of a glacier within a similar frequency than the one used in the present study is dominated by water flow, and particularly subglacial water flow. The authors of Nanni et al., (2021) localize, thanks to a 100-seismometers array, the sources of such noise and show that at the beginning of the melt season the sources are distributed while they become localized as the melt season progresses. The authors also show that in both cases the source of seismic noise is the turbulent water flow, and as soon as the subglacial drainage system becomes efficient, the seismic noise is dominated by the turbulent water flow originating from a 'channelized' drainage system.

In the present study, we do not need to differentiate if the noise comes from a conduit and not a distributed system. First, because we are in a setup with basal sediments, which make the conduit/cavity terminology and distinction less relevant in comparison to Nanni et al., (2021)’s study where they worked on an hard-bed glacier. Second, because we look at the seasonal evolution of the hydraulic properties, while Nanni et al.,

(2021) investigated bi-daily changes in subglacial hydraulic properties in order to depict the transition between inefficient and efficient drainage system. Third, and maybe the most relevant point for our study, whether the noise originates from a distributed or localized drainage system has little to none effect on our interpretations, indeed at such frequency band and with our seismic network geometry we average the hydraulic properties of one to two times the ice thickness (300 to 600 m, see Nanni et al. (2020)) and therefore we are investigating the spatially averaged evolution of subglacial water flow properties.

Given these points and the reviewers comments we have clarified our manuscript and we have modified the main and the supplementary material to make this aspect more clear.

From line 197, we have added:

- [...] which can originate, on glaciers, from both an efficient/localized and inefficient/distributed subglacial drainage system (Nanni et al., 2021).
- [...] relative changes in spatially averaged subglacial hydraulic radius and pressure gradient [...] We indeed states that:
 - The hydraulic radius expresses the influence of wall friction for a given cross-section (Methods, Eq. A11), such that an increase in relative hydraulic radius (R ; size-coded in Fig. 3) can be interpreted as an increase in subglacial drainage capacity (Nanni et al., 2020). The hydraulic gradient combines the gradient of subglacial water pressure and the bed slope (Methods, Eq. A12), such that an increase in the relative hydraulic gradient (S ; color-coded in Fig. 3) can be interpreted as a pressurization of the subglacial drainage system (Gimbert et al., 2016).

In the supplementary material we have added from line 633: - As shown by Nanni et al. (2021), seismic noise within the selected frequency (i.e., 5 - 10 Hz) is dominated by turbulent water flow. The authors show that such flow happens in the distributed/inefficient drainage system, and that when a localized/efficient drainage system develops, the seismic noise generated by the turbulent water flow within such a system tends to dominates the signal. In our study, we do not make any assumptions on the geometry of the subglacial drainage system, but rather investigate the evolution of the capacity and pressure conditions through parameters than can be seen as bulk parameters representing the conditions in the subglacial drainage system averaged over one to two ice thickness, i.e., 300 to 600 m (Nanni et al., 2020).

Minor points

Line 35. "This uncertainty is particularly due to the erratic nature of dynamic instabilities" reads awkwardly

AR: We have rephrased as:

This uncertainty is particularly due to the unpredictable nature of dynamic instabilities (Kochtitzky, Copland, Van Wychen, Hugonnet, et al., 2022), such as glacier surges, i.e., glacier-wide acceleration over one or several orders of magnitude.

Line 141-142. The main thing that is being evoked in affecting variable A_s within the Weertman sliding law is the water pressure so why not state that here. Note Weertman has no real physical meaning at this point as he did not consider the role of water pressure or soft beds.

AR: We first start by stating (starting from line 158) that we use A_s as a bulk basal sliding parameter; the higher A_s the more the glacier slips (Fig. 2c). This parameter accounts for various potential basal processes, such as sediment deformation and basal hydrology effects.

We agree with the reviewer that A_s has no real physical meaning in the Weertman sliding law, and this is why we focus on this paragraph (starting from line 158) on dis-

cussing the evolution of A_s in term of basal friction conditions and not in term of basal water pressure. Our goal here, has well detailed by the reviewer in his previous comment, is to use an agnostic sliding law to disentangle first between changes in geometry and changes in basal conditions. We then use (from line 172) a water pressure-dependent sliding law, in order to discuss the evolution of the basal water pressure, as we use a sliding law that is expressly dependent on basal water pressure through the effective pressure N .

Line 304-305. Stevens et al., (2024) saw this same sort of response at a glacier in Canada and their findings are pretty closely aligned with your interpretations

AR: We thank the reviewer for this suggestion, we have now included a reference to Stevens et al., (2024) in addition to the reference to Graff et al., 2021.

Fig A1. add x label (years)

AR: We have made this change.

Fig A3. again P_a-2 which does not make sense for a choice of $n=3$.

AR: We have made this change, and answered in detail to this comment in the first major comment of this reviewer.

Line 541-544. It would be clearer to just explicitly state the events were not located

AR: We have precised this on the main text (line 222):

[...] We did not localize these icequakes [...]

Line 574-575. State how abundant (what percentage of the overall time) the times are with negative correlations

AR: Here we do not define clear thresholds or time windows over which we define positive and negative correlation, we rather qualitatively show that both behavior are observed, and that there is not a unique trend between seismic rate and seismic noise.

Line 586. cannot to can no

AR: We have changed this.

Reviewer #2 - William David Harcourt**Key results**

This paper documents a rich observational time series of surface and subsurface dynamics during the recent acceleration of Kongsvegen. Recent evidence has suggested that Kongsvegen is entering the active phase of a surge, yet it is unclear what behaviour will manifest as a result. The paper analyses seismic records, satellite elevation change data, runoff modelling and model outputs of ice dynamics to infer that a hydro-mechanical feedback is the cause of the recent acceleration. They identify the formation of a surge bulge that has led to surface steepening and changes in driving stress and basal shear stress. Observations of bed conditions at the initiation of a surge are rare and so this paper presents a unique insight into the processes that drive these initial conditions. This has wider implications for ice flow modelling and the potential for long-term changes as a result of warmer air temperatures, surface melting, and hydrological feedbacks on ice flow.

General Comments

I am not a modeller, hence my comments focus mostly on the observational methods and their interpretation. I found this paper to be well-written and presents an intriguing time series that is extremely difficult to acquire. For that, I commend the authors and their work. There are a few areas that I think require a bit further clarification before publication:

1. The land/marine terminating boundary condition is not mentioned at all the text. I see in response to Reviewer 2 the authors suggest that a change from land- to marine-terminating has a negligible impact on the hydrologic pressure gradient, but what about frontal stresses and calving? One would expect a marine boundary (and Kongsvegen's detachment from neighbouring Kronebreen) to be more sensitive to surface steepening due to a reduction in frontal buttressing. How could this process impact the interpretation of a hydro-mechanical feedback enhancing the observed ice flow acceleration, particularly nearer the terminus?

AR: We agree with the reviewer that the dynamics of the glacier front may have some influence on surface elevation changes close to the glacier terminus. Land-terminating glaciers typically slow down towards the glacier front and surface elevation changes are therefore typically dominated by patterns of surface mass balance. In contrast, many marine-terminating glaciers accelerate towards the front, due to reduced flow resistance, especially during periods of frontal retreat. Accordingly, surface elevation changes close to the terminus may be in variable degree influenced, if not dominated by ice dynamical processes. Nevertheless, these effects are site specific and cannot be generalized to all marine-terminating glaciers; therefore we do not cover this in our discussion. Due to the generally slow surface speed of Kongsvegen in its lower part (less than a few meters per years, see Fig 1d), the dynamical component in surface elevation changes of this particular glacier is very small; this view is further corroborated by (Nuth et al., 2012). In addition, in the present study, we focus on the acceleration in the upper part of Kongsvegen, more than 12 km upglacier from its terminus. Furthermore, in our analysis we account for potential longitudinal stress coupling using observed profiles of strain rate. Therefore, even if Kongsvegen would exhibit a partial acceleration towards its marine terminus, our results would not be affected by this effect.

In order to clarify this aspect we have added the following precisions in our manuscript: Section 2.1:

- Kongsvegen glacier [...] has since exhibited a slow and stable flow until 2014 (Bouchayer et al., 2024), , with mass loss dominated by surface mass balance and little to none influence of calving (Nuth et al., 2012)

- In 2023, the lower part of the glacier (locations 1 to 3; km 0–6 Fig. 1d) exhibited low U_s , with melt-season (April–September) averages below 20 m yr^{-1} . There, thinning reached up to 30 m between 2005 and 2023, mostly related to changes in surface mass balance (Nuth et al., 2012).

2. Related to the above point, I would ask the authors to emphasise throughout the manuscript that surface steepening is induced by ablation at lower elevations and accumulation at higher elevation. As currently written, the text implies the whole of the glacier surface has ablated at a constant rate that is not spatially variable.

AR: In the case of Kongsvegen, the changes in surface mass balance are closely related to change in surface elevation (Nuth et al., 2012). In the manuscript we do not state that the whole glacier has ablated at a constant rate, and in Section 2.1 we detail that the changes in surface elevation are not constant over the glacier:

- From 2005 to 2014, Kongsvegen exhibited uniformly low glacier surface velocity (U_s) with annual values below 5 m yr^{-1} alongside noticeable thinning, particularly near the glacier front.

- While thinning remains most pronounced near the glacier front, the acceleration occurs in the upper part of the ablation zone (locations 7, 8; km 12–14).

- Conversely, in the upper part of the glacier (locations 7 to 8; km 12–14) and near the ELA, U_s exhibited a ten-fold increase over the same period, with melt-season averages up to 80 m yr^{-1} in 2023. This area experienced a thickening since 2019, particularly pronounced in 2023 (up to +10 m), while the area immediately up-glacier (km 14–16) displayed minor thinning ($> -5 \text{ m}$) and less pronounced acceleration.

To clarify this point, we have added the following sentence in this section:

- [...] thinning reached up to 30 m between 2005 and 2023, mostly related to changes in surface mass balance (? , ?).

3. In the discussion, the authors claim that surface steepening has led to an increase in driving stress and point to Fig. 2a. This appears to not be supported as the driving stress data (d) in Fig. 2a below, where it is 6 km is essentially the same. The only regions where I see driving stress changes are in the central region, which might imply the surge bulge is moving downglacier.

AR: In Fig. 2a we observe an increase in driving stress d between 2005 and 2023, particularly from km 3 to km 12. We have now specified in the main text that the increase in driving stress is between km 3 at km 12 and not from from km 0 to km 12. The increase in driving stress at km 6 that the reviewer is pointing out, is not related to the bulge, as the bulge is observed between km 10 and 14, as shown in Fig 1e. The increase in driving stress at km 6 is rather caused by a steepening of the glacier surface, as we describe at the beginning of the Discussion section:

- Since 2005, and particularly since 2014, Kongsvegen experienced surface steepening in most of the ablation area (km 0–12; Fig. 1e,f), leading to an increase in driving stress (km 3–12; Fig. 2a).

What do the authors believe the velocity increases near the terminus might be related to; on the evidence provided, it seems less likely to be related to the surge bulge - changes in the terminus e.g. to marine-terminating?

The most significant acceleration is observed in the upper part of the ablation zone (km 8 to 16), where velocities have increased by more than a factor of 10, and not near the glacier front, where velocities have mostly stagnated from 2005 to 2019. In this study, our main focus is the acceleration of the upper part of the glacier, which likely impacts also the lower part of the glacier. As stated earlier in an answer to the same reviewer, the contact with the ocean has no significant effect on our analysis. We have clarified the fact that our main focus is on the upper part of the glacier as follows in Section 3:

- we jointly discuss our results to explain the recent acceleration of the upper part of the glacier in the context of a self-amplifying *hydro-mechanical feedback*.
- Concomitantly, surface velocities in this region increased by more than one order of magnitude (Fig. 1d), in strong contrast to the slowly moving lower part of the glacier where we observe only a minor acceleration.

We have also clarified in the introduction:

- [we conduct] in-situ measurements during the initiation of a partial acceleration of the upper part of Kongsvegen glacier (78.8°N, 13.3°E, Fig. 1a) that may develop into a glacier-wide surge (Bouchayer et al., 2024; Mannerfelt et al., 2025).

And in the methods section, we have clarified that the, small, increase in surface velocity near the glacier front is more likely due to the propagation of the surge bulge rather than to the reduction in the frontal stresses. Indeed, if it was related to a reduction in frontal stress we would have observed an increase in along-flow tensile stress near the front, while we observe here the opposite with an increase in along-flow compressive stress :

Line 150:

- Near the glacier front, the increase in along-flow compression (km 0–3, Fig. 2a) indicates a stagnation of the glacier front, which suggests that there is little to none reduction in frontal stresses due to frontal ablation.

4. As mentioned in my comments below, it would be useful to include velocity data as a time series in Fig. 3 and maybe also Fig. 4. This would help to demonstrate the seasonal coupling between hydrologic changes and velocities.

AR: Timeseries of glacier surface velocity are already shown in Fig. 4 (top rows), for the years 2018, 2019, 2022 and 2023 for two sites, one in the upper part of the glacier, the other one on the lower part. In Fig. 3, we include the timeseries of runoff, with highlighted periods that corresponds between Fig 3 and 4 in order to help the reader analyses the link between runoff and glacier velocity. Complete timeseries of surface velocity are then shown in Supplementary figure A6.

Minor Comments

L12: Maybe worth defining it as a surge e.g. ‘instability, termed a surge’

AR: We have rephrased as:

[...] Sudden glacier acceleration and instability, e.g. surges, strongly influence glacier ice loss. [...]

L14: Change ‘role’ to ‘process’

AR: Here we refer to the role of sudden glacier acceleration on glacier ice loss. We refer to the involved processes earlier in the sentence.

[...] However, a lack of in-situ observations of the involved processes hampers our ability to understand, quantify and model such a role. [...]

L14: ‘analysis of’

AR: We have changed this following the reviewer’s suggestion.

L18: State location of thinning e.g. in the ablation zone or up to 8 km from the terminus.

AR: Here we remain broad as we are in the abstract and provide further details in the main text.

L21: I suggest stating that this is a hydro-mechanical feedback somewhere here.

AR: We indeed use this term in the main text, especially in the discussion, but we would rather keep it simple in the abstract without adding extra terminology.

L25: ‘mass loss from glaciers and ice caps’

AR: We have rephrased as suggested.

L43: Could also cite McMillan et al. (2014) here.

AR: we thank the reviewer for this suggestion, yet we keep the reference to Dunse et al., 2015 as this study quantifies the mass loss during the Austfonna surge.

L44: Suggest change to ‘Surges initiate due to a build up of internal energy (Benn et al., 2019)

AR: We thank the reviewer for this suggestion, yet rather refer to the original studies on surge behavior.

L55-58: It can be this, but also ablation thinning in the terminus region causing acceleration - this should also be acknowledged (e.g. Sevestre et al., 2018; surge propagation from the terminus of Aavatsmarkbreen)

AR: In Sevestre et al., 2018, the surge of Aavatsmarkbreen is explained by surface meltwater reaching previously inaccessible areas of the bed and leading to decreased basal friction. This is permitted by increased surface crevassing. In the main text, we indeed refer to such a mechanism when we state:

[...] increased surface ablation has been suggested to promote surging behavior (Dunse et al., 2014; Sevestre et al., 2018; Nuth et al., 2019). This later mechanism is caused by surface meltwater reaching previously inefficient parts of the subglacial drainage system, resulting in high basal water pressure, which in turn reduces the ice-bed mechanical coupling and favors glacier sliding (Liboutry, 1968). [...]

We do not specifically talk if this surge initiation takes place at the glacier front and propagate upglacier or vice-versa. We have also provided more details on this aspect in the previous comments from this reviewer.

L64: ‘gap by conducting’

AR: We have rephrased as suggested.

L65: ‘the partial acceleration’ - I don’t think ‘initiation’ is needed here as it implies you think it will evolve into a full surge, which at this stage I don’t think we can be sure of.

AR: We agree with the reviewer that this might not evolve into a full surge, this is why we refer to [...] *the initiation of a partial acceleration of Kongsvegen glacier [...]. Here we study the initiation of this partial acceleration, and its subsequent propagation.* By using the term initiation, we do not imply that the acceleration will become glacier-wide, we rather refer to the fact that we are investigating the processes leading to the initiation of this local acceleration and not just the acceleration by itself. Hence, we kept the original phrasing.

L67: ‘where the mean annual air temperature has increased’

AR: We have rephrased as suggested.

L95: Notable thinning? About 3 m since 2005 (considering just the blue lines)? This seems quite small.

AR: We have changed to: Noticeable.

L150: ‘To disentangle these effects,’

AR: The term ‘Second’ refers to the fact that at line 158 we state [First, we reconstruct ...], because we first start with the Weertman approach and then we use the Schoof approach.

L192-204: It is not clear to me what you are looking to find from the icequakes activity, I assume information on basal sliding dynamics, Could you clarify this in the text?

AR: On line 222 we detail that we use these icequakes to investigate surface crevassing, basal crevassing and stick-slip events:

[...] *we differentiated between near-surface icequake, typically attributed to surface crevassing (Walter et al., 2008) and near-bed icequakes, typically attributed to basal crevass-*

ing and stick-slip events (Gräff et al., 2021) by comparing records from co-located surface and borehole seismometers [...].

To make it more clear we have rephrased the introductory sentence (line 192) as:
 [...] *To unravel the processes responsible for this pressurization of the ice-bed interface, we used seismic measurements to reconstruct the evolutions of subglacial drainage system properties (Fig.3) as well as of the occurrence of icequakes to investigate the evolution of crevasses and basal friction (Fig. 4).* [...]

L232: Should reference Figure 1 here when mentioning U_s , and when referenced in subsequent paragraphs. There is a lot going on both Figs. 3 and 4. Is it possible to state U_s values anywhere for easy reference? From figure A2 you appear to have a time series of velocities, could you add a separate panel to each of the columns displaying this?

AR: Here we refer to the seasonal evolution of U_s , which is only displayed on Figure 4 and not in Figure 1 where we display the averaged values. We refer to both Figure 3 and 4 since we refer to the seasonal evolution of runoff (Figure 3) and U_s (Figure 4). In Figure 4 we have already put the U_s values on top of each column for the four seasons. We have shown only for two location to make the reading easy and then we have added the full time series in A2.

L270: The driving stress changes are small in the ablation zone, as you have stated on Lines 125-127, and then on Lines 127-129 you state there has been a small change in the central area. Theoretically surface steepening should lead to an increase in driving stress, this is currently not supported by your observations.

AR: We agree with the reviewer that in Figure 2 and the related description we describe small changes in the driving stress. As we describe in Eq.1, the driving stress $\tau_d = \rho_i g H \sin(\alpha)$, where ρ_i is the density of ice, g the gravitational acceleration, H the glacier thickness, and α the glacier surface slope. In this equation, we can indeed see that surface steepening leads to increasing driving stress if the thickness is constant. In our case we both observe a surface steepening and a thinning of the glacier. Hence, surface steepening might not lead to increase in driving stress if this is counterbalanced by ice thinning.

L295: This goes back to my previous comment on Figs. 3 and 4 - I would like to see the hydrologic data inferred from the seismic data displayed alongside the velocity data to fully capture this relationship - seasonal velocity data is not displayed anywhere in the paper otherwise.

AR: We understand this suggestion. The hydrological data inferred from the seismic data are shown in Figure 3, together with the runoff evolution and the velocity data are shown in Figure 4 together with the icequakes investigation. Given the different dataset and the spatial and temporal coverage, we made such choices of the representation to allow the reader to have a clear spatial and temporal view of our different datasets.

Complete timeseries of surface velocity are then shown in Supplementary Figure A6, as well as complete timeseries of runoff (Fig. A1) and timeseries of seismic power are shown in Figure A6.

L333: ‘succeeded in simulating’

AR: We have rephrased as suggested.

L361: ‘On the one hand’

AR: We have rephrased as suggested.

L381: Not sure I understand this point - most, if not all, surge-type glaciers in Svalbard are polythermal, so the hydro-mechanical feedback as escribed here only applies to polythermal / temperate glaciers?

AR: In the case of Kongsvegen glacier, the basal condition are temperate, which is not the case of all surge-glaciers in Svalbard, where bed conditions may also be cold. What we state is that the process we describe here does not involve the presence of a cold bed,

or the transition for a cold to temperate bed, and therefore extends to glaciers outside of the High Arctic with only temperate bed conditions.

L384: ‘Although Kongsvegen surged in 1948 (Melvold Hagen, 1998) and has slowly been accumulating mass’

AR: We have rephrased as suggested.

L388: ‘The combination of stronger’

AR: We have rephrased as suggested.

L401-418: What about observations needed to better understand this process? Can we rely on satellite measurements or do we need a few ‘super sites’ of geophysical investigations like in this study to parameterise this process in models?

AR: This is a very open question, and this is within such a perspective that we have framed our study.

L409: Remove first both

AR: We have rephrased as suggested.

Figure A.10: What year was the imagery from? Is there evidence that these crevasses have appeared in the timeframe of the acceleration? Or could they have always been present?

AR: As stated in the legend: the imagery is from August 1 2020: [...]Aerial view on the Kongsvegen on August 1, 2020 [...]. The fact that we see, especially in panels d, f, the relics of the supraglacial channel suggest that these crevasses are recent.

Reviewer #3 - Dual anonymous**General**

In this paper, the authors analyze a surge of Kongsvegen glacier in Svalbard. They integrate in-situ observations over the last two decades (GNSS, borehole and surface seismometers) with runoff simulations, and remotely sensed surface-elevation changes. The data are interesting and their interpretation is supported. However, it is not clear that this paper adds significantly to the body of literature about surges. I think that the paper should be published somewhere, but I am not sure that Nature Communications is the right venue. I think the Journal of Glaciology would be a better fit.

AR: We thank the reviewer for its review. We detail below our answers to its specific comments.

Major Comments

I have a few significant comments

1. The interpretation of the water pressure and velocity involves circular reasoning. The water pressure is calculated from the velocity, therefore concluding that high water pressure is driving the acceleration, without providing additional information/models/data is seemingly a logical fallacy. I understand that the seismic data is corroboratory but it is not a direct inference.

AR: We thank the reviewer for highlighting this aspect, and we would like to emphasize the structure of our reasoning. First, we observe an increase in the glacier velocity (observed from GNSS), then we add independent measurements (e.g. ice geometry changes) and theorize through 2 different sliding laws how a local acceleration may be composed of different parts (changes in driving stress, basal shear stress, longitudinal stress and basal conditions). By doing so we, qualitatively, evaluate the relative contribution of the ice-dynamics components and we infer the water pressure from the residuals. We find that the influence of the ice-dynamics components is much smaller than that of water pressure, leading us to the conclusion that this was the main driver. We then further corroborate this finding using the seismic measurements that add another independent source of information, to propose a series of processes responsible for such an acceleration.

This structure shows that our interpretation does not involve a circular reasoning, but rather a step-by-step-constructed reasoning supporting by different independent observations and a clear physical framework.

2. This study is focused on the interpretation of a single glacier surge, which is limited in scope. I know that we can, and need to, learn from single locations, but I would hope that there is something larger that could be drawn from the observations. I am not sure how to generalize the results to surges, glaciers in general, or to ice sheets

AR: The way to learn from single locations/events is to deduce the underlying processes and discuss how these may unfold at a different location/ in a different case. In our study, we provide an unprecedentedly rich observational account for a feedback that has been hypothesized from observations at other glaciers. It is therefore clear that this effect applies to other glaciers as well. Additionally, the significance of the processes we observe on a global scale still remains to be determined by future studies, as the goal of our study is to highlight and document a mechanisms that was yet mostly observed on cold-based glaciers, while we show that it can also happen on temperate based glacier, therefore expanding its significance. In the discussion, we explain this and provide references. We have reworked our discussion to better include such a perspective.

Minor Comments

I have numerous small comments 1. line 55: 'later' should be 'latter', I believe

AR: We have rephrased as suggested.

2. Figure 1(b): this is nit picky but the red box should be around the outside of this box, and then the study area could be circled within the inner box.

AR: We have changed as suggested.

3. Figure 5: I am confused by the blue color here - I now see that it is the glacier and not water, but labels could clarify.

AR: We thank the reviewer for this suggestion, we have updated the label accordingly.

4. line 474: there is a missing space between (1957)' and law. Moreover, I wouldn't suggest using the citations as possessive.

AR: We have rephrased as suggested.

Reviewer #2

The authors have sufficiently addressed the comments I made in my previous statement with the new version of the text. I have noted a few small edits below

Other comments—

Line 56 therefore key for our improving our $\dot{\gamma}$ therefore key for improving our

Line 174 : “Here, we assume a uniform distribution of $C = 0.3$ ” but on line 499 it says $C=0.5$. update these to be the same

484: extra space after tau l

In a few instances some of the citations have first initials that should be removed

AR: We thank the reviewer for the final edits, and we have revised the manuscript accordingly.

Reviewer #4

Nanni et al. use modelling and seismic observations to study the dynamics of Kongsvegen since 2000, focusing on the upper regions of the glacier where a surge bulge appears to have formed. Kongsvegen is a known surge-type glacier and the thickening observed here suggests it is currently building mass towards a surge, which is backed-up by evidence of velocity increases in the same region. Modelling shows that basal shear stress has increased across most of the glacier, including in the region of velocity increases, leading to a loss of basal friction and therefore enabling the glacier to slide. The authors interpret these findings in the context of a hydro-mechanical feedback, which has been suggested previously for surges in Svalbard (e.g. Basin-3, Austfonna), whereby water can more readily penetrate to the bed through crevasses which have opened up due to the increase in along-flow tensile stress. The observations represent a very detailed record of observations during the surge initiation and the modelling enables suitable interpretation of the data in the context of glaciological theory. As presented, I believe the paper can be published as is, although I provide some very minor technical comments below.

Thank-you to the authors for a very nice study!

AR: We thank the reviewer for the positive review.

Technical Corrections (References to line (L) numbers in preprint)

L75: ‘increasing basal water pressure’

L76: ‘facilitates’

L111: ‘little to no’

L336: ‘an increase’

L356: ‘remain localized or (ii) evolve into ’

L374: ‘Kongsvegen’s current glacier dynamics’

L395: ‘combination of’

L397: ‘triggering of’

L445: Reference for this? I assume this is taken from the RGI7.0.

L637: ‘In our study’

L644: ‘elastic waves’

L706: Just to be clear this is $DEM = DEM_{2014} \frac{dh}{dt}$, where dh/dt is from Hugonnet et al. (2021)

AR: We have revised the manuscript following the above editing suggestions. We have added the reference to Nuth et al., 2013 instead of RGI7 and we have clarified the reference to Hugonnet et al., (2021).

Reviewer #5

This paper compiles a large amount of field observations as well as inferences from force balance and a sliding law. As before, I still do not follow the logic of the paper. The authors' response was helpful in understanding their argument. However, those insights do not appear in the main text from what I can tell. If the paper is going forward, I suggest a clearer statement about the logic flow and a description of the issues that arise in the logical.

AR: We thank the reviewer for his comment. We have now improved the following sentences at the end of the introduction:

In this study, we investigate, through a multi-method and multi-temporal analysis, how climatic and glacier-specific drivers have worked in tandem to initiate Kongsvegen's current instability. First we diagnose the dynamical consequences of climate-driven geometric changes using on-ice GNSS measurements and geodetically-derived elevation changes. **In particular, we evaluate the relative contributions of the ice-dynamics components and of the changes in basal water pressure condition to the observed acceleration.** Second, we identify the surface and subglacial processes responsible for these changes using passive seismic records collected at the glacier surface and near its bed. Third, we interpret Kongsvegen's recent acceleration as the result of a climatic-induced, self-amplifying mechanism we refer to as a *hydro-mechanical feedback*. Finally, we discuss how this mechanism may contribute to a broader glacier-wide destabilization and contextualize the implications of our findings within a global perspective.

Review of Manuscript NCOMMS-23-63486B: Observed positive feedback between surface ablation and crevasse formation drives glacier acceleration and potential surge

William David Harcourt, University of Aberdeen

February 9, 2025

Key Results

This paper documents a rich observational time series of surface and subsurface dynamics during the recent acceleration of Kongsvegen. Recent evidence has suggested that Kongsvegen is entering the active phase of a surge, yet it is unclear what behaviour will manifest as a result. The paper analyses seismic records, satellite elevation change data, runoff modelling and model outputs of ice dynamics to infer that a hydro-mechanical feedback is the cause of the recent acceleration. They identify the formation of a surge bulge that has led to surface steepening and changes in driving stress and basal shear stress. Observations of bed conditions at the initiation of a surge are rare and so this paper presents a unique insight into the processes that drive these initial conditions. This has wider implications for ice flow modelling and the potential for long-term changes as a result of warmer air temperatures, surface melting, and hydrological feedbacks on ice flow.

General Comments

I am not a modeller, hence my comments focus mostly on the observational methods and their interpretation. I found this paper to be well-written and presents an intriguing time series that is extremely difficult to acquire. For that, I commend the authors and their work. There are a few areas that I think require a bit further clarification before publication:

1. The land/marine terminating boundary condition is not mentioned at all the text. I see in response to Reviewer #2 the authors suggest that a change from land- to marine-terminating has a negligible impact on the hydrologic pressure gradient, but what about frontal stresses and calving? One would expect a marine boundary (and Kongsvegen's detachment from neighbouring Kronebreen) to be more sensitive to surface steepening due to a reduction in frontal buttressing. How could this process impact the interpretation of a hydro-mechanical feedback enhancing the observed ice flow acceleration, particularly nearer the terminus?
2. Related to the above point, I would ask the authors to emphasise throughout the manuscript that surface steepening is induced by ablation at lower elevations and accumulation at higher elevation. As currently written, the text implies the whole of the glacier surface has ablated at a constant rate that is not spatially variable.
3. In the discussion, the authors claim that surface steepening has led to an increase in driving stress and point to Fig. 2a. This appears to not be supported as the driving stress data (τ_d)

in Fig. 2a below, where it is 6 km is essentially the same. The only regions where I see driving stress changes are in the central region, which might imply the surge bulge is moving downglacier. What do the authors believe the velocity increases near the terminus might be related to; on the evidence provided, it seems less likely to be related to the surge bulge - changes in the terminus e.g. to marine-terminating?

4. As mentioned in my comments below, it would be useful to include velocity data as a time series in Fig. 3 and maybe also Fig. 4. This would help to demonstrate the seasonal coupling between hydrologic changes and velocities.

Technical Corrections (References to line (L) numbers in preprint)

L12: Maybe worth defining it as a surge e.g. 'instability, termed a surge'

L14: Change 'role' to 'process'

L14: 'analysis of'

L18: State location of thinning e.g. in the ablation zone or up to 8 km from the terminus.

L21: I suggest stating that this is a hydro-mechanical feedback somewhere here.

L25: 'mass loss from glaciers and ice caps'

L43: Could also cite McMillan et al. (2014) here.

L44: Suggest change to 'Surges initiate due to a build up of internal energy (Benn et al., 2019), often'

L55-58: It can be this, but also ablation thinning in the terminus region causing acceleration - this should also be acknowledged (e.g. Sevestre et al., 2018; surge propagation from the terminus of Aavatsmarkbreen)

L64: 'gap by conducting'

L65: 'the partial acceleration' - I don't think 'initiation' is needed here as it implies you think it will evolve into a full surge, which at this stage I don't think we can be sure of.

L67: 'where the mean annual air temperature has increased'

L95: Notable thinning? About 3 m since 2005 (considering just the blue lines)? This seems quite small.

L150: 'To disentangle these effects,'

L192-204: It is not clear to me what you are looking to find from the icequakes activity, I assume information on basal sliding dynamics, Could you clarify this in the text?

L232: Should reference Figure 1 here when mentioning U_s , and when referenced in subsequent paragraphs. There is a lot going on both Figs. 3 and 4. Is it possible to state U_s values anywhere for easy reference? From figure A2 you appear to have a time series of velocities, could you add a separate panel to each of the columns displaying this?

L270: The driving stress changes are small in the ablation zone, as you have stated on Lines 125-127, and then on Lines 127-129 you state there has been a small change in the central area. Theoretically surface steepening should lead to an increase in driving stress, this is currently not supported by your observations.

L295: This goes back to my previous comment on Figs. 3 and 4 - I would like to see the hydrologic data inferred from the seismic data displayed alongside the velocity data to fully capture this relationship - seasonal velocity data is not displayed anywhere in the paper otherwise.

L333: 'succeeded in simulating'

L361: 'On the one hand'

L381: Not sure I understand this point - most, if not all, surge-type glaciers in Svalbard are polythermal, so the hydro-mechanical feedback as escribed here only applies to polythermal / temperate glaciers?

L384: 'Although Kongsvegen surged in 1948 (Melvold & Hagen, 1998) and has slowly been accumulating mass'

L388: 'The combination of stronger'

L401-418: What about observations needed to better understand this process? Can we rely on satellite measurements or do we need a few 'super sites' of geophysical investigations like in this study to paramterise this process in models?

L409: Remove first both

Figure A.10: What year was the imagery from? Is there evidence that these crevasses have appeared in the timeframe of the acceleration? Or could they have always been present?

Review: “Observed positive feedback between surface ablation and crevasse formation drives glacier acceleration and potential surge”

dual anonymous peer review

Submitted to the *Nature Communications*

1 General

In this paper, the authors analyze a surge of Kongsvegen glacier in Svalbard. They integrate in-situ observations over the last two decades (GNSS, borehole and surface seismometers) with runoff simulations, and remotely sensed surface-elevation changes. The data are interesting and their interpretation is supported. However, it is not clear that this paper adds significantly to the body of literature about surges. I think that the paper should be published somewhere, but I am not sure that *Nature Communications* is the right venue. I think the *Journal of Glaciology* would be a better fit.

2 Remarks

I have a few significant comments

1. The interpretation of the water pressure and velocity involves circular reasoning. The water pressure is calculated from the velocity, therefore concluding that high water pressure is driving the acceleration, without providing additional information/models/data is seemingly a logical fallacy. I understand that the seismic data is corroboratory but it is not a direct inference.
2. This study is focused on the interpretation of a single glacier surge, which is limited in scope. I know that we can, and need to, learn from single locations, but I would hope that there is something larger that could be drawn from the observations. I am not sure how to generalize the results to surges, glaciers in general, or to ice sheets.

3 Specific comments

I have numerous small comments

1. line 55: ‘later’ should be ‘latter’, I believe
2. Figure 1(b): this is nit picky but the red box should be around the outside of this box, and then the study area could be circled within the inner box.
3. Figure 5: I am confused by the blue color here - I now see that it is the glacier and not water, but labels could clarify.
4. line 474: there is a missing space between (1957)’ and law. Moreover, I wouldn’t suggest using the citations as possessive.

Review of Manuscript NCOMMS-23-63486C:
Observed positive feedback between surface ablation and crevasse
formation drives glacier acceleration and potential surge

William David Harcourt, University of Aberdeen

July 22, 2025

Comments

Nanni et al. use modelling and seismic observations to study the dynamics of Kongsvegen since 2000, focusing on the upper regions of the glacier where a surge bulge appears to have formed. Kongsvegen is a known surge-type glacier and the thickening observed here suggests it is currently building mass towards a surge, which is backed-up by evidence of velocity increases in the same region. Modelling shows that basal shear stress has increased across most of the glacier, including in the region of velocity increases, leading to a loss of basal friction and therefore enabling the glacier to slide. The authors interpret these findings in the context of a hydro-mechanical feedback, which has been suggested previously for surges in Svalbard (e.g. Basin-3, Austfonna), whereby water can more readily penetrate to the bed through crevasses which have opened up due to the increase in along-flow tensile stress.

The observations represent a very detailed record of observations during the surge initiation and the modelling enables suitable interpretation of the data in the context of glaciological theory. As presented, I believe the paper can be published as is, although I provide some very minor technical comments below.

Thank-you to the authors for a very nice study!

Technical Corrections (References to line (L) numbers in preprint)

L75: 'increasing basal water pressure'

L76: 'facilitates'

L111: 'little to no'

L336: 'an increase'

L356: 'remain localized or (ii) evolve into'

L374: 'Kongsvegen's current glacier dynamics'

L395: 'combination of'

L397: 'triggering of'

L445: Reference for this? I assume this is taken from the RGI7.0.

L637: ‘In our study’

L644: ‘elastic waves’

L706: Just to be clear this is $DEM = DEM_{2014} - dh/dt$, where dh/dt is from Hugonot et al. (2021)?